# The glucose-sensing transcription factor MLX balances metabolism and stress to suppress apoptosis and maintain spermatogenesis

Patrick A. Carroll[1], Brian W. Freie[1], Pei Feng Cheng[1], Sivakanthan Kasinathan[1], Haiwei Gu[2], Theresa Hedrich[1], James A. Dowdle[3], Vivek Venkataramani[4], Vijay Ramani[5], Xiaoying Wu[1], Daniel Raftery[2], Jay Shendure[5,6,7], Donald E. Ayer[8], Charles H. Muller[9], Robert N. Eisenman[1] *

1 Basic Sciences Division, Fred Hutchinson Cancer Research Center, Seattle, Washington, United States of America, 2 Department of Anesthesiology and Pain Medicine, University of Washington, Seattle, Washington, United States of America, 3 Molecular Biology Program, Memorial Sloan Kettering Cancer Center, New York, New York, United States of America, 4 Institute of Pathology, University Medical Center Göttingen, Göttingen, Germany, 5 Department of Genome Sciences, University of Washington, Seattle, Washington, United States of America, 6 Howard Hughes Medical Institute, Seattle, Washington, United States of America, 7 Brotman Baty Institute for Precision Medicine, Seattle, Washington, United States of America, 8 Huntsman Cancer Institute, Department of Oncological Sciences, University of Utah, Salt Lake City, Utah, United States of America, 9 Male Fertility Lab, Department of Urology, University of Washington, Seattle, Washington, United States of America

* eisenman@fredhutch.org

**Data Availability Statement:** The underlying data for the RNA-seq and ChIP-seq datasets can be found at GSE165820.

## Abstract

Male germ cell (GC) production is a metabolically driven and apoptosis-prone process. Here, we show that the glucose-sensing transcription factor (TF) MAX-Like protein X (MLX) and its binding partner MondoA are both required for male fertility in the mouse, as well as survival of human tumor cells derived from the male germ line. Loss of *Mlx* results in altered metabolism as well as activation of multiple stress pathways and GC apoptosis in the testes. This is concomitant with dysregulation of the expression of male-specific GC transcripts and proteins. Our genomic and functional analyses identify loci directly bound by MLX involved in these processes, including metabolic targets, obligate components of male-specific GC development, and apoptotic effectors. These in vivo and in vitro studies implicate MLX and other members of the proximal MYC network, such as MNT, in regulation of metabolism and differentiation, as well as in suppression of intrinsic and extrinsic death signaling pathways in both spermatogenesis and male germ cell tumors (MGCTs).

## Introduction

The MYC/MAX/MXD network plays a critical role in both development and tumorigenesis as major mediators of transcriptional regulation of growth, metabolism, proliferation, apoptosis, and differentiation (for reviews, see [1–3]). This network is comprised of basic helix–loop–helix–leucine zipper (bHLHLZ) transcription factors (TFs) generally associated with activation (MYC) or repression (MXD) that compete for an obligate heterodimerization partner (MAX)

**Funding:** PAC, PFC, BWF, TH, XW and RNE were supported by a grant from the National Cancer Institute at the National Institutes of Health USA (https://www.cancer.gov/grants-training): R35 CA231989 (to RNE). PAC was also supported by a postdoctoral fellowship T32 CA009657. Scientific Computing Infrastructure at the Fred Hutchinson Cancer Research Center was funded by an ORIP grant from the National Institutes of Health: S10OD028685 (https://orip.nih.gov). SK was supported by a fellowship from the National Institutes of Health USA F30CA186458 (https://www.cancer.gov/grants-training). HG and DR were supported by Cancer Center Support Grant, National Institutes of Health P30 CA015704 (https://www.cancer.gov/grants-training). JS is an Investigator of the Howard Hughes Medical Institute (https://www.hhmi.org/programs). DEA is supported by National Institutes of Health R01 GM055668 (https://www.cancer.gov/grants-training) The funders had no role in study design, data collection and analysis, decision to publish, or preparation of the manuscript.

**Competing interests:** I have read the journal's policy and the authors of this manuscript have the following competing interests: RNE is a member of the Scientific Advisory Boards of Kronos Bio Inc. and Shenogen Pharma, Beijing. There is no overlap between the research presented in this manuscript with the products or methods related to these companies.

**Abbreviations:** bHLHLZ, basic helix–loop–helix–leucine zipper; CHEA, ChIP set enrichment analysis; ChIP-Seq, chromatin immunoprecipitation and sequencing; ChREBP, carbohydrate response element binding protein; DEG, differentially expressed gene; DSP, daily sperm production; E-box, Enhancer box; ESI, electrospray ionization; GC, germ cell; GSEA, gene set enrichment analysis; HILIC, hydrophilic interaction chromatography; IACUC, institutional animal care and use committee; IF, immunofluorescence; JAK, cytokine-activated Janus kinase; KO, knockout; LC–MS/MS, liquid chromatography with tandem mass spectrometry; LFC, log fold change; MDF, Modified Davidson's Fluid; MGCT, male germ cell tumor; MLX, MAX-Like protein X; MLXIP, MLX-interacting protein; MRM, multiple reaction monitoring; MSigDB, Molecular Signature Database; NFκB, nuclear factor kappa B; OA, oleic acid; OAT, oligoasthenoteratozoospermia; PCA, principal component analysis; PLS-DA, partial least squares discriminant analysis; QC, quality control; RNA-seq, RNA sequencing; Sc, spermatocyte; scRNA-seq, single-cell RNA sequencing; sFASL, soluble FAS ligand; siRNA, small interfering RNA;

in order to bind DNA and influence expression of shared target genes [4]. Typically, MYC-MAX responds to mitogenic signals to activate Enhancer box (E-box)-containing promoters, whereas MXD-MAX responds to the loss of mitogenic signals or differentiation cues to repress the same targets. This allows the network to balance proliferative cues with cell cycle entry and exit.

The MAX-centered network exists within a larger network, containing MAX-Like protein X (MLX), MondoA (also known as MLX-interacting protein, MLXIP), and carbohydrate response element binding protein (ChREBP, also known as MondoB and MLXIPL; reviewed in [5,6]). MLX heterodimerizes with a subset of MXD proteins as well as the glucose-sensing MondoA and ChREBP but is unable to heterodimerize with either MAX or MYC. ChREBP-MLX [7] and MondoA-MLX heterodimers are major regulators of glucose-responsive transcription in vitro and in vivo [8–11], and MLX function has been linked to the response to metabolic stress in multiple organisms [12–15]. Genetic ablation of MYC [16], MAX [17], and MNT [18] (the most ubiquitously expressed MXD family member) results in embryonic or perinatal lethality (in the case of MNT); however, loss of MondoA [19] or ChREBP does not interfere with overt development [7]. We previously demonstrated an obligate role for the MLX arm of the network in promoting survival of a wide range of tumor cells with deregulated MYC by facilitating metabolic reprogramming and suppressing apoptosis. However, cells expressing endogenously regulated MYC were found to tolerate MondoA or MLX loss [20].

Here, we report the phenotype associated with loss of MLX during normal murine development. As with deletion of either *Mlxip* (*MondoA*) or *Mlxipl* (*Chrebp*), loss of *Mlx* is not detrimental to normal embryonic development or organismal viability. However, all male homozygous null animals (MLX^KO) exhibit complete sterility with a dramatic increase in apoptosis of germ cells (GCs). Many of these phenomena are recapitulated by Sertoli cell–specific deletion of *Mlx*, directly implicating MLX in the normal function of this male-specific stromal cell. We link this phenotype to a broad integrated transcriptional program mediated by MLX within the MYC network that facilitates metabolism and directly suppresses apoptosis.

## Results

### Homozygous deletion of *Mlx* is developmentally tolerated but results in male-only infertility

To examine potential developmental roles of *Mlx* in the mouse, we generated a targeting construct for deletion of exons 3 to 6 of the murine *Mlx-α* isoform encoding the bHLHLZ region required for dimerization and DNA binding (Fig 1A, S1A Fig). Upon constitutive heterozygous deletion of *Mlx*, we were able to obtain both *Mlx*^+/− males and females, indicative of developmental haploinsufficiency, as reported for heterozygous deletions of *Myc* or *Max*. However, upon mating of heterozygous males and females, we were surprised to discover that, unlike complete deletion of *Myc* or *Max*, which results in embryonic lethality, homozygous loss of *Mlx* is well tolerated, resulting in offspring at the expected mendelian frequencies (Fig 1B). *Mlx* null (MLX^KO) mice of both sexes were indistinguishable from wild-type (WT) mice, lived a normal life span, and exhibited normal behavior, including copulation. Similar to *Mlx*^+/− (HET) mice of either sex, *Mlx*^−/− females were able to breed successfully, whereas all *Mlx*^−/− males were infertile (Fig 1C).

Coincident with the loss of fertility, MLX^KO testis and epididymis tissue were disorganized, and we observed markedly reduced populations of GCs and spermatozoa compared with WT mice. The weights of both testis and epididymis were significantly decreased in MLX^KO adult males (Fig 1D and 1E) despite normal body weight (Fig 1F). MLX^KO testes frequently

Spg, spermatogonia; SSC, spermatogonial stem cell; St, spermatid; STAT, signal transducer and activator of transcription; T, testosterone; TCA, tricarboxylic acid; TF, transcription factor; TNFα, tumor necrosis factor alpha; TSS, transcription start site; VIP, variable importance to projection; WB, western blot; WT, wild-type.

exhibited abnormal and acellular seminiferous tubules (marked with blue asterisks in Fig 1G), and MLX$^{KO}$ epididymides contained decreased numbers of spermatozoa, which displayed highly abnormal morphologies relative to WT including malformed heads and abnormal mid-piece and tail structures (S1C Fig), as well as populations of cells with an immature appearance compared with WT (Fig 1G). As shown in Fig 1H, the GC identity of these immature appearing cells within the MLX$^{KO}$ epididymis was confirmed by staining with the pan-GC cytoplasmic marker DDX4 (also known as VASA). In WT mice, DDX4 is only detected at low levels in cells within the epididymis due to removal by phagocytosis of spermatid (St) cytoplasts or residual bodies [21] possessing this marker during the transition from round to elongated St (Fig 1H). These epididymal histological phenotypes of MLX$^{KO}$ were not present in *Mlx* heterozygous animals (S1B Fig) and were observed with varying severity from age P51 onward (S1D Fig). Note the combination of both abnormal spermatozoa and immature round cells in the epididymis of MLX$^{KO}$ animals even at this young age (S1E Fig).

## Infertility of MLX$^{KO}$ male mice originates in the testes

To begin investigating a role for MLX in male fertility, we first determined the extent of MLX expression in the testes using immunofluorescent antibody staining. Fig 2A and 2B shows staining with anti-MLX throughout the testis, including GCs, Sertoli cells, interstitial cells, as well as in the lining of the epididymis, where spermatozoa mature and gain motility. The observed staining is specific, as no signal is detected from secondary antibody or from the same tissue derived from our MLX$^{KO}$ mice and stained with anti-MLX. We note that MLX expression in general appears to decrease with differentiation into the lumen of the seminiferous tubule (Fig 2B). Such widespread expression of MLX is consistent with other available data from human testes confirming the presence of this transcript in multiple testes cell types. The Human Protein Atlas [22] (http://www.proteinatlas.org) shows similar staining for MLX in human testis and epididymis, and single-cell RNA sequencing (scRNA-seq) of human testicular cell populations [23] shows *MLX* specifically enriched in Sertoli and primitive GCs (S2A Fig) and co-expressed with *MLXIP* (encoding MondoA) (S2B Fig). These data support a conserved role for MLX in human testes.

To understand the basis for the testicular phenotype in the MLX$^{KO}$ mice, we quantified aspects of the cellular biology of the testicular and epididymal GCs. Enumeration of daily sperm production (DSP) rate revealed both a significant decrease in DSP (Fig 3A), as well as diminished output of mature sperm to the epididymis in the MLX$^{KO}$ compared with WT mice (Fig 3B), consistent with decreased production in the testis. While the majority of WT cauda spermatozoa were motile and had a normal appearance, spermatozoa that reached the cauda epididymis in MLX$^{KO}$ animals exhibited both a lack of progressive motility (Fig 3C) and abnormal morphology (Fig 3D), as shown previously in (S1C and S1E Fig). These features are consistent with the clinical symptoms of oligoasthenoteratozoospermia (OAT) as defined by decreased sperm number, lack of motility, and altered morphology of the sperm [24].

To determine whether a defect in androgen production correlated with the loss of fertility in MLX$^{KO}$ males, we quantified serum testosterone (T) between WT and MLX$^{KO}$ animals. Consistent with the lack of a change in the size of the seminal vesicle (Fig 3E), a T responsive tissue, we did not detect a change in serum T levels between genotypes (Fig 3F). By contrast, alterations in levels of important metabolites were detected upon liquid chromatography with tandem mass spectrometry (LC–MS/MS) metabolomic analysis of serum from WT versus MLX$^{KO}$ males. Partial least squares discriminant analysis (PLS-DA; S3A Fig), carried out using MetaboAnalyst [25], revealed multiple metabolites with a variable importance to projection (VIP) score of greater than 1. The top 20 are shown in S3B Fig, and the heat map in S3C Fig

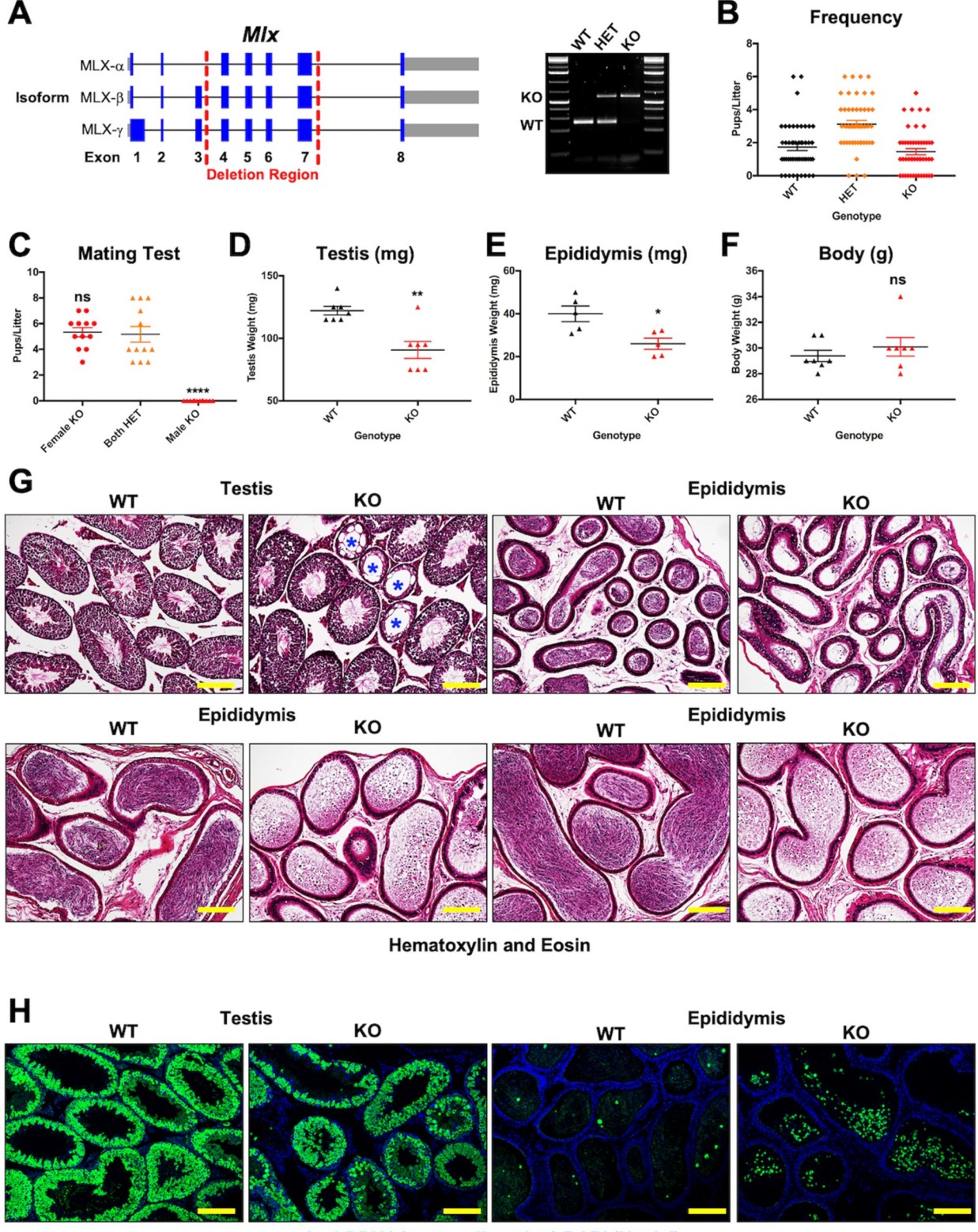

**Fig 1. Targeting strategy and initial reproductive characterization of the MLX[KO] mice.** (**A**) Cloning strategy for deletion of exons 3 to 6, encoding the DNA-binding domain, of murine Mlx-α with PCR products of the 3 potential genotypes WT, HET, and KO shown to the right. (**B**) Pups/litter from mating tests between MLX HET males and females (*N* = 48). (**C**) Mating test results from crossing the indicated sex and MLX

genotype mice ($N$ = 12). (**D**) Testis weight of WT versus MLX$^{KO}$ mice ($N$ = 7). (**E**) Epididymis weight of WT and MLX$^{KO}$ mice ($N$ = 5). (**F**) Body weight of WT and MLX$^{KO}$ mice ($N$ = 7). (**G**) Histological analysis of WT and MLX$^{KO}$ testis and epididymis stained with hematoxylin and eosin; asterisks mark degenerated seminiferous tubules (100×, scale bar = 400 uM). (**H**) IF analysis of WT and MLX$^{KO}$ testis and epididymis stained for the indicated protein (100×, scale bar = 400 uM). Shown for all is the mean with SEM with $p$-values shown from paired $t$ test for groups of 2 and ANOVA for groups of 3 or more (* $p < 0.05$, ** $p < 0.01$, *** $p < 0.001$, *** $p < 0.0001$). The underlying data for Fig 1B–1F can be found in S1 Data. HET, heterozygous; IF, immunofluorescence; KO, knockout; MLX, MAX-Like protein X; WT, wild-type.

with the complete log fold change (LFC)-normalized dataset is included as S1 Table. Ribose-5-phosphate, pyruvate, and lactate were increased in MLX$^{KO}$ serum (S3B and S3C Fig), indicative of enhanced whole body glycolysis. MLX$^{KO}$ mice also exhibited alterations to metabolite levels associated with amino acid oxidation, such as decreased valine and a buildup of 2 downstream metabolites, 3-amino-isobutyrate and 2-hydroxy-isovaleric acid. Augmented glycolysis and alterations to oxidative substrates have also been reported in mice lacking MondoA or treated with a chemical inhibitor of MondoA [19] [10]. MondoA-MLX heterodimers are known to act through their downstream target, TXNIP, which, in turn, suppresses glycolysis [8]. Taken together, these results indicate that deletion of *Mlx* leads to a change, not in the production of T, but in whole body metabolism consistent with loss of MondoA-MLX activity, as well as alterations in normal testicular and epididymal tissue homeostasis. A model depicting changes in mitochondrial oxidative substrates upon MLX loss is shown in S3D Fig, suggesting a switch from glucose to amino acid oxidation.

As MLX functions as a TF in concert with its heterodimeric binding partners, MLX interacting proteins, we hypothesized that such a binding partner could also be required for male fertility. Since previous deletion of *Mlxipl* (encoding ChREBP) did not affect fertility [7], we gauged the requirement of *Mlxip* (encoding MondoA) for male fertility. As shown in Fig 3G, deletion of the gene encoding MondoA results in male-only infertility. Interestingly, in contrast to the MLX$^{KO}$ spermatozoa, which appeared abnormal, the spermatozoa from MondoA$^{KO}$ mice appear normal and are produced at normal number (S3E Fig) but are completely nonmotile (asthenospermic) (Fig 3H). This both supports a direct transcriptional requirement for MondoA-MLX activity in male fertility and suggests that MLX has functions independent of MondoA in the context of spermatogenesis.

Given the phenotypic differences between the MondoA and the MLX deleted mice, we sought to specifically determine the stage of the defect in spermatogenesis in the MLX$^{KO}$ mice. Staining for phospho-Histone H3ser10 (to detect mitotic and meiotic cells) indicated that GCs from both WT and MLX$^{KO}$ testes could undergo successful meiosis in the testis (Fig 3I). WT testis showed the expected stage-specific expression of γH2AX (marker of meiosis and DNA damage) decreasing with differentiation and present only in rare epididymal cells (most likely round St shed from the testes). By contrast, the MLX$^{KO}$ tissue display disrupted expression and the shedding of immature, γH2AX+ cells into the epididymis (Fig 3J). We also observed ectopic epididymal staining for the pan-GC marker DDX4 (see Fig 1H), which is normally only detected at low levels during the transition from round to elongated St.

As the cytoplasmic cell markers DDX4 and γH2AX are normally lost upon completion of meiosis, we tested whether testicular and epididymal cells from MLX$^{KO}$ mice were arresting in meiosis. All stages of meiosis (1, 2, and 4n DNA content) were observed in the testis and epididymis of both WT and MLX$^{KO}$ mice (Fig 3K), indicating that the cells transiting to the epididymis of MLX$^{KO}$ mice, although having significantly reduced total cell numbers compared with WT (Fig 3B), were postmeiotic (predominantly 1n). However, the MLX$^{KO}$ epididymal population displayed increased sub-1n DNA content, indicative of DNA fragmentation (Fig 3K), consistent with the observed γH2AX expression. In conclusion, MLX$^{KO}$ males exhibit OAT, with decreased testicular St production accompanied by impaired transition from round

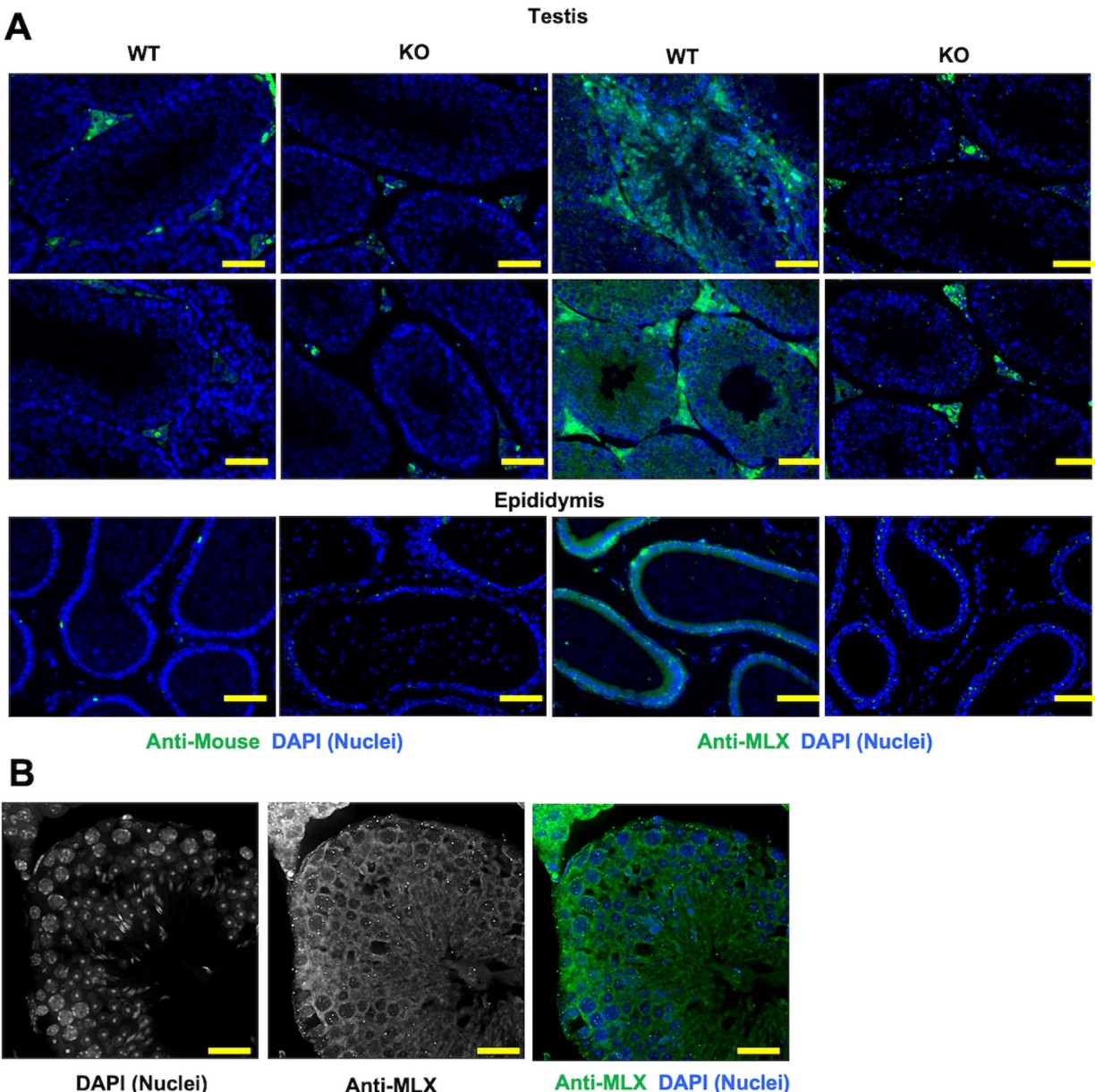

**Fig 2. Immunostaining for MLX in WT and MLX$^{KO}$ male reproductive tissue.** (**A**) IF analysis of WT versus MLX$^{KO}$ testis and epididymis stained with either secondary alone, or with anti-MLX then secondary antibody (200×, scale bar = 200 uM). (**B**) IF analysis of WT testis tissue stained with anti-MLX showing the single DAPI and MLX, as well as combined, channels (400×, scale bar = 100 uM). IF, immunofluorescence; KO, knockout; MLX, MAX-Like protein X; WT, wild-type.

to elongated St morphology. These immature postmeiotic cells maintain meiosis and stress markers and are shed to the epididymis where they appear to undergo apoptosis.

## MLX deletion in Sertoli cells partially phenocopies the whole body deletion

Because Sertoli cells are known to play a critical role in the maintenance of the seminiferous epithelium to support GC development and prevent shedding of immature cells, we asked whether Sertoli cells were affected by MLX loss. We stained WT and MLX$^{KO}$ testes with anti-

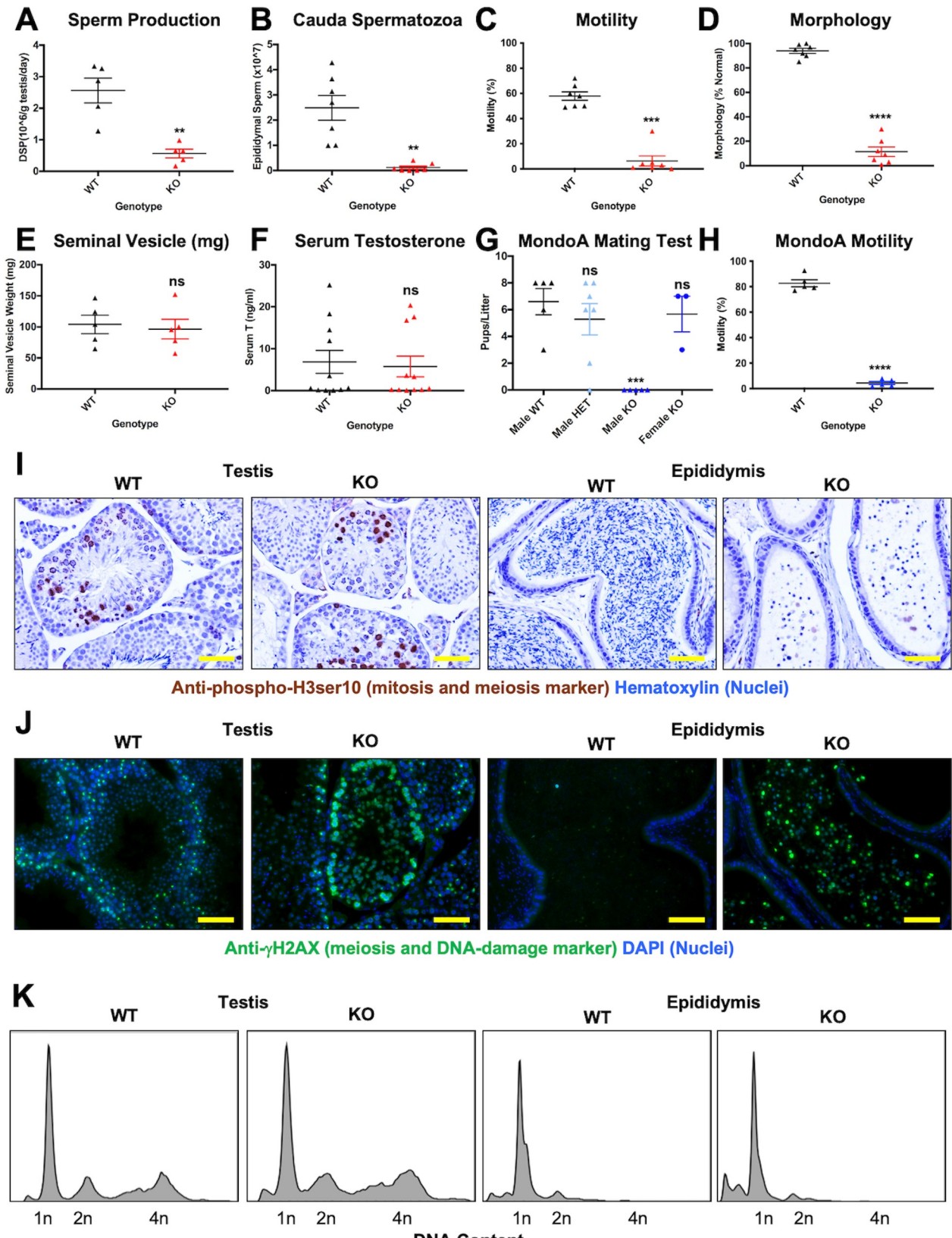

**A** Sperm Production
**B** Cauda Spermatozoa
**C** Motility
**D** Morphology
**E** Seminal Vesicle (mg)
**F** Serum Testosterone
**G** MondoA Mating Test
**H** MondoA Motility

**I** Testis / Epididymis

Anti-phospho-H3ser10 (mitosis and meiosis marker) Hematoxylin (Nuclei)

**J** Testis / Epididymis

Anti-γH2AX (meiosis and DNA-damage marker) DAPI (Nuclei)

**K** Testis / Epididymis

DNA Content

**Fig 3. Fertility traits of WT, MLX$^{KO}$, and MondoA$^{KO}$ mice.** (A–F) Comparison of WT and MLX$^{KO}$ mice: (**A**) DSP (*N* = 5 mice). (**B**) Epididymal sperm counts (*N* = 7 mice). (**C**) Percent progressive motility of cauda epididymal sperm (*N* = 7 mice). (**D**) Percent normal morphology of epididymal sperm (*N* = 7 mice). (**E**) Seminal vesicle weight (*N* = 5 mice). (**F**) Serum T (*N* = 11 mice). (**G**) Mating test results from crossing the indicated sex and MondoA genotype mice (*N* = 3 to 6). (**H**) Percent progressive motility of cauda epididymal sperm from WT and MondoA$^{KO}$ mice (*N* = 5). (**I**) IHC analysis of WT and MLX$^{KO}$ testis and epididymis stained for the indicated protein (200×, scale bar = 200 uM). (**J**) IF analysis of testis and epididymis stained for the indicated protein (200×, scale bar = 200 uM). (**K**) Flow cytometry analysis of single cell suspension from testis and epididymis of WT versus MLX$^{KO}$ mice stained for DNA content. Shown for all is the mean with SEM with *p*-values shown from paired *t* test for groups of 2 and ANOVA for groups of 3 or more (* $p < 0.05$, ** $p < 0.01$, *** $p < 0.001$, *** $p < 0.0001$). The underlying data for Fig 3A–3H can be found in S1 Data, and the underlying data for Fig 3K can be found in S2 Data. DSP, daily sperm production; HET, heterozygous; IF, immunofluorescence; IHC, immunohistochemistry; KO, knockout; MLX, MAX-Like protein X; T, testosterone; WT, wild-type.

SOX9, which is present in Sertoli cell nuclei in a highly characteristic pattern at the periphery of seminal vesicle tubules. We found that SOX9 staining was essentially unaffected in seminal tubules from either WT or MLX$^{KO}$ testes with normal GC abundance (Fig 4A). However, as noted above, a fraction of MLX$^{KO}$ seminiferous tubules appeared to be largely devoid of GCs yet maintained an outer ring of peripheral epithelial cells (Fig 1G). These acellular tubules in the MLX$^{KO}$ testes exhibited an increase in the density and frequency of peripheral SOX9 positive cells, possibly indicative of Sertoli cell dysfunction (Fig 4A).

To determine whether Sertoli cell function was dependent on MLX, we compared our whole body deletion of *Mlx* with the effects of a targeted Sertoli cell *Mlx* deletion using Amh-Cre [26] in *Mlx*$^{fl/fl}$ mice. We found that, similar to MLX$^{KO}$ males, the *Amh-Cre; Mlx*$^{fl/fl}$ mice exhibited male infertility (Fig 4B). However, unlike whole body knockout (KO) males with decreased testis and epididymis weight, there was no change in the weight of any of the tissues examined (Fig 4C–4F). As observed in the whole body KO, *Amh-Cre; Mlx*$^{fl/fl}$ males exhibited defects in spermatogenesis, including decreased cauda epididymal sperm content resulting from diminished DSP, as well as loss of motility and abnormal morphology (Fig 4G–4J). Importantly, however, unlike whole body deletion of *Mlx*, the relative DSP rate was significantly higher in the Amh-Cre+ mice than in our whole body MLX$^{KO}$ (Fig 4K). Furthermore, mice bearing Sertoli-specific deletion of MLX did not possess acellular seminiferous tubules that are prevalent (accounting for about 10% of the total) in the whole body MLX$^{KO}$ mice (Fig 4L, S4B Fig; quantified in S4C Fig). These findings further confirm that the deleterious effects of MLX loss on fertility can originate within the testis and are consistent with the essential linkage between Sertoli cell function and spermatogenesis, including coupled glucose and lipid metabolism (reviewed in [27,28]). However, the differences in sperm production rate and lack of acellular tubules suggest that the MLX loss in Sertoli cells only partially phenocopies the constitutive loss of MLX.

In order to gauge where MondoA-MLX transcriptional activity is present, we stained WT and MLX$^{KO}$ testes for SOX9 and TXNIP, as a proxy. As shown in S4D Fig, TXNIP is widely expressed in the testes and greatly decreased with deletion of MLX. While *TXNIP* is highly expressed in human Sertoli cells (S4E Fig), the protein decreases in both GCs and Sertoli cells in the MLX$^{KO}$, suggestive of MLX activity in both compartments. Consistent with this, another MondoA-MLX target, *ARRDC4*, is highly expressed in GC compared with Sertoli cells (S4F Fig), and *Arrdc4* was recently demonstrated to support murine sperm maturation in vivo [29]. Given the widespread expression of MLX in the testis (Fig 2A and 2B), we reasoned that MLX loss is likely to have a cell autonomous effect on GC differentiation and spermiogenesis and decided to focus our further investigation on the whole body MLX$^{KO}$.

## Expression profiling reveals decreased spermatogenesis, altered metabolism, and increased stress in MLX$^{KO}$ testes

To gauge the altered transcriptome of MLX$^{KO}$ tissue, we used RNA sequencing (RNA-seq) to profile the RNA of whole testes from age-matched littermates of WT verified fertile breeders

versus constitutive MLX$^{KO}$ males ($n$ = 3 pairs). As shown in Fig 5A, principal component analysis (PCA) of these samples indicates that they group according to genotype. We identified 4,688 differentially expressed genes (DEGs) upon loss of MLX (2,282 up and 2,406 down) (Fig 5B, S2 Table). Gene set enrichment analysis (GSEA) indicated enrichment for only the Spermatogenesis Hallmark Gene Set in the WT testes, which correlates with normal spermatogenesis in these mice (Fig 5C). However, compared with WT, the MLX$^{KO}$ tissue is enriched for gene expression signatures related to multiple metabolic pathways including fatty acid metabolism, glycolysis, and oxidative phosphorylation. We also noted enrichment for stress pathways in the MLX$^{KO}$ tissue, including inflammatory and interferon responses, tumor necrosis factor alpha (TNFα), nuclear factor kappa B (NFκB), cytokine-activated Janus kinase (JAK)-signal transducer and activator of transcription (STAT) and WNT signaling, and apoptosis (Fig 5C, S3 Table).

Spermatogenesis is a highly choreographed developmental program that can be separated into 3 broad categories enriched in 3 GC types (spermatogonia (Spg), spermatocyte (Sc), and spermatid (St)), which undergo, respectively, self-renewal/mitosis, meiosis, and spermiogenesis (see schematic in Fig 5D). These cellular states are lineage specified through the activities of specific TFs and their targets, several of which are listed in Fig 5D. Similar to GSEA, Enrichr analysis [33,34] for ChIP set enrichment analysis (CHEA) was employed to identify TFs associated with DEGs from WT compared with MLX$^{KO}$ testes. Most noteworthy is that genes down-regulated in MLX$^{KO}$ cells are significantly associated with loss of CREM and MYBL1 (required for spermiogenesis [35] and meiosis [36], respectively), while up-regulated genes were associated with the more primitive spermatogonial TFs DMRT1 [37], OCT4 [38], and MYC [39] (S5A Fig). This suggests an incomplete block in normal spermatogenesis in MLX$^{KO}$ GCs with a loss of late markers and an accumulation of more primitive markers of GC differentiation.

We next asked whether genes whose expression is modulated by MLX deletion overlap with those controlled by key transcriptional regulators of spermatogenesis. We prepared volcano plots using previously reported DMRT1-bound [30] (Fig 5G) and CREM-bound [31] (Fig 5H) genes from mouse testis that overlap with DEGs determined by our RNA-seq data. DMRT1 is known to balance mitosis and meiosis induction [40], and the trend toward up-regulation of these targets (217 up versus 117 down) is consistent with the entry of these cells into meiosis, as further evidenced by the presence of cells with 1N DNA content (see Fig 3K). The decreased expression of both MYBL1 and CREM targets is likely to be associated with stress during meiosis and spermiogenesis, respectively. Indeed, while the majority of CREM-bound targets are down-regulated in our MLX$^{KO}$ RNA-seq analysis (1031 down versus 534 up), we find that a large fraction of DEGs in MLX$^{KO}$ testes correspond to up-regulated genes previously shown to be linked to deletion of *Crem*, but not all of which are directly bound by CREM (622/652 up-regulated genes, as opposed to 640/898 down-regulated genes) [32] (CREM-LFC, Fig 5I). Our data implicate MLX in DMRT1- and CREM-regulated pathways critical for mammalian spermatogenesis.

In order to highlight significantly altered transcripts from the GSEA/CHEA categories identified, we generated heat maps for the top 100 down-regulated and up-regulated DEGs (Fig 5E and 5F). These include loss of both primitive SSC and Spg markers such as *Eomes*, *Lhx1*, and *Prom1* (purple tabs), as well as the more differentiated St markers *Tnp1*, *Tnp2*, *Prm1*, and *Prm2* (green tabs). We observe a concerted gain of apoptosis markers and effectors (red tabs), which include *Timp1*, *Casp4*, *Atf3*, *Igfbp3*, *Casp12*, *and Fas* up-regulation (Fig 5F) as well as loss of *Txnip*, a proapoptotic protein that is MLX dependent for expression [8] (S4D Fig). Intriguingly, we also see the up-regulation of many genes associated with Sertoli dysfunction and stress, including the induction of the feminizing signaling molecule *Wnt4* and stress markers known to be Sertoli enriched [41].

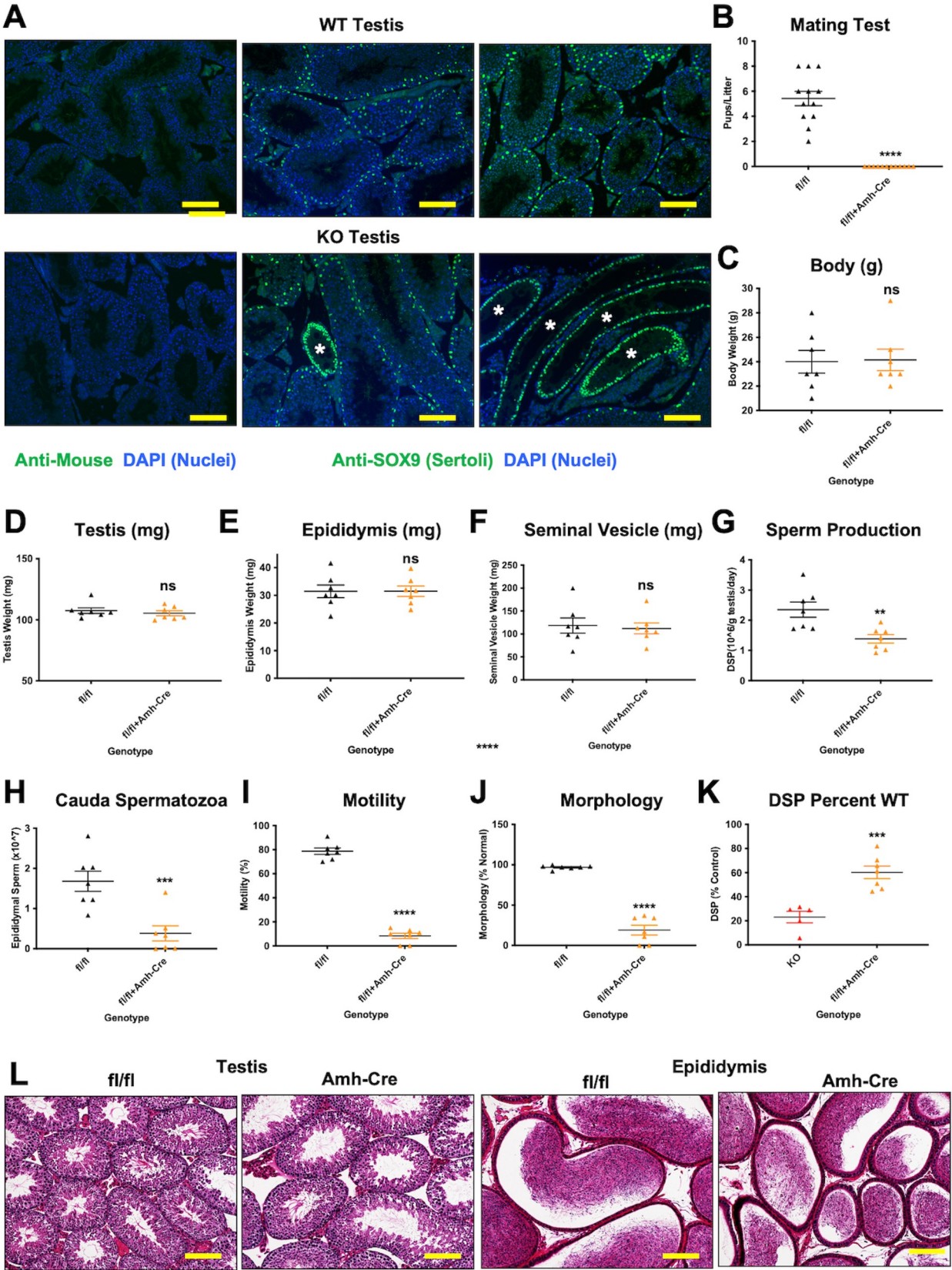

**Fig 4. Analysis of targeted MLX deletion in Sertoli cells.** (**A**) IF for the Sertoli cell marker SOX9 in WT and MLX$^{KO}$ testes with acellular tubules indicated with asterisks (100×, scale bar = 400 uM). (**B**) Pups/litter of mating tests between MLX$^{fl/fl}$ males and Amh-Cre+ males with WT females ($N = 7$). (**C**) Body weight of MLX$^{fl/fl}$ males compared with Amh-Cre+ littermates ($N = 7$). (**D**) Testis weight of MLX$^{fl/fl}$ males compared with Amh-Cre+ littermates ($N = 7$). (**E**) Epididymis weight of MLX$^{fl/fl}$ males compared with Amh-Cre+ littermates ($N = 7$). (**F**) Seminal Vesicle weight of MLX$^{fl/fl}$ males compared with Amh-Cre+ littermates ($N = 7$). (**G**) DSP rate of MLX$^{fl/fl}$ males compared with Amh-Cre+ littermates ($N = 7$). (**H**) Cauda epididymal sperm counts from MLX$^{fl/fl}$ males compared with Amh-Cre+ littermates ($N = 7$). (**I**) Percent of cauda epididymal sperm of MLX$^{fl/fl}$ males compared with Amh-Cre+ littermates exhibiting progressive motility ($N = 7$). (**J**) Percent normal morphology of cauda epididymal sperm of MLX$^{fl/fl}$ males compared with Amh-Cre+ littermates ($N = 7$). (**K**) Comparison of the decrease in DSP rate between WT to MLX$^{KO}$ and MLX$^{fl/fl}$ to MLX$^{fl/fl}$ with Amh-Cre ($N = 5$–7). (**L**) Histological analysis of testis and epididymis of 6-month-old MLX$^{fl/fl}$ males compared with Amh-Cre+ littermates stained with hematoxylin and eosin (100×, scale bar = 400 uM). Note the lack of tubules with loss of GCs. Shown for all panels is the mean with SEM with $p$-values shown from paired $t$ test, except for 4K that was an unpaired $t$ test (* $p < 0.05$, ** $p < 0.01$, *** $p < 0.001$, *** $p < 0.0001$). The underlying data for Fig 4B–4K can be found in S1 Data. DSP, daily sperm production; GC, germ cell; IF, immunofluorescence; KO, knockout; MLX, MAX-Like protein X; WT, wild-type.

Immunoblotting of whole testes confirmed that a subset of these differentially expressed transcripts are also altered at the protein level. As shown in S5B Fig, upon MLX loss, we observed decreased expression of both MLX dimerization partners MondoA and ChREBP, decreased expression of the known MondoA- or ChREBP-MLX target gene TXNIP, as well as decreased EOMES protein. However, consistent with GSEA enrichment for apoptosis and inflammation categories, we observed increased FAS (death receptor CD95) expression (S5B Fig) as well as reactivity with anti-mouse IgG HRP, indicative of resident immune cells, present in both WT and MLX$^{KO}$, but increased in MLX$^{KO}$ testes (S5C Fig).

We employed immunofluorescence on WT and MLX$^{KO}$ testes to ascertain in situ, which populations of cells are altered upon loss of MLX. While both WT and MLX$^{KO}$ testes stain positive for the pan-GC marker DDX4, the mature St marker PGK2 is significantly decreased in MLX$^{KO}$ tissue (Fig 5J). The residual PGK signal detected in Fig 5J is due to PGK1, present in interstitial and somatic cells. DDX4 expression confirms the GC fate of a large fraction of the testis cells. This, and the enrichment for DMRT1 targets among up-regulated DEGs (Fig 5G), suggests that the proliferative Spg population is still present in MLX$^{KO}$ testes. Consistent with this, the proliferation mark Ki67 is relatively unchanged in MLX$^{KO}$ compared with WT tissue, aside from tubules that lose cells (S5D Fig). Taken together with the loss of markers such as PGK2, our data suggest that loss of MLX causes defects in the transition from Sc to St.

In contrast to the loss of PGK2, there is widespread staining for both the inflammation marker TIMP1 (S5E Fig) and FAS (Fig 5K, S5F Fig) in the seminiferous tubules and the interstitium of the testes, as well as the shed GCs present in the epididymides of MLX$^{KO}$ mice. A higher magnification image is presented for FAS staining in Fig 5K, demonstrating that the round, immature St that are both positive for DDX4 and γH2AX are indeed also expressing high levels of FAS compared with isotype control staining. This indicates that the loss of GCs associated with MLX$^{KO}$ tissue results from activation of the extrinsic cell death pathway via FAS death receptor signaling coincident with inhibition of differentiation.

## Gene expression in fractionated testes cell populations

We next fractionated the testicular tissue to remove interstitial (stromal and immune) cells from the seminiferous tubules (comprised predominantly of Spg, Sc, and St as well as Sertoli cells) in order to assess protein expression. In comparison with WT, fractionated MLX$^{KO}$ tubule cells show complete loss of MLX and strongly decreased expression of both MLX dimerization partners, MondoA and ChREBP (Fig 6A). Surprisingly, these cells also exhibit moderately decreased expression of the immature spermatogonial stem cell (SSC) markers MYCN, MAX, and OCT4, with no change in the expression of the MYC-antagonist MNT (Fig 6A). We had also noted diminished expression of the SSC marker EOMES in whole testes (S5B Fig). Importantly, small interfering RNA (siRNA) against MLX resulted in similar

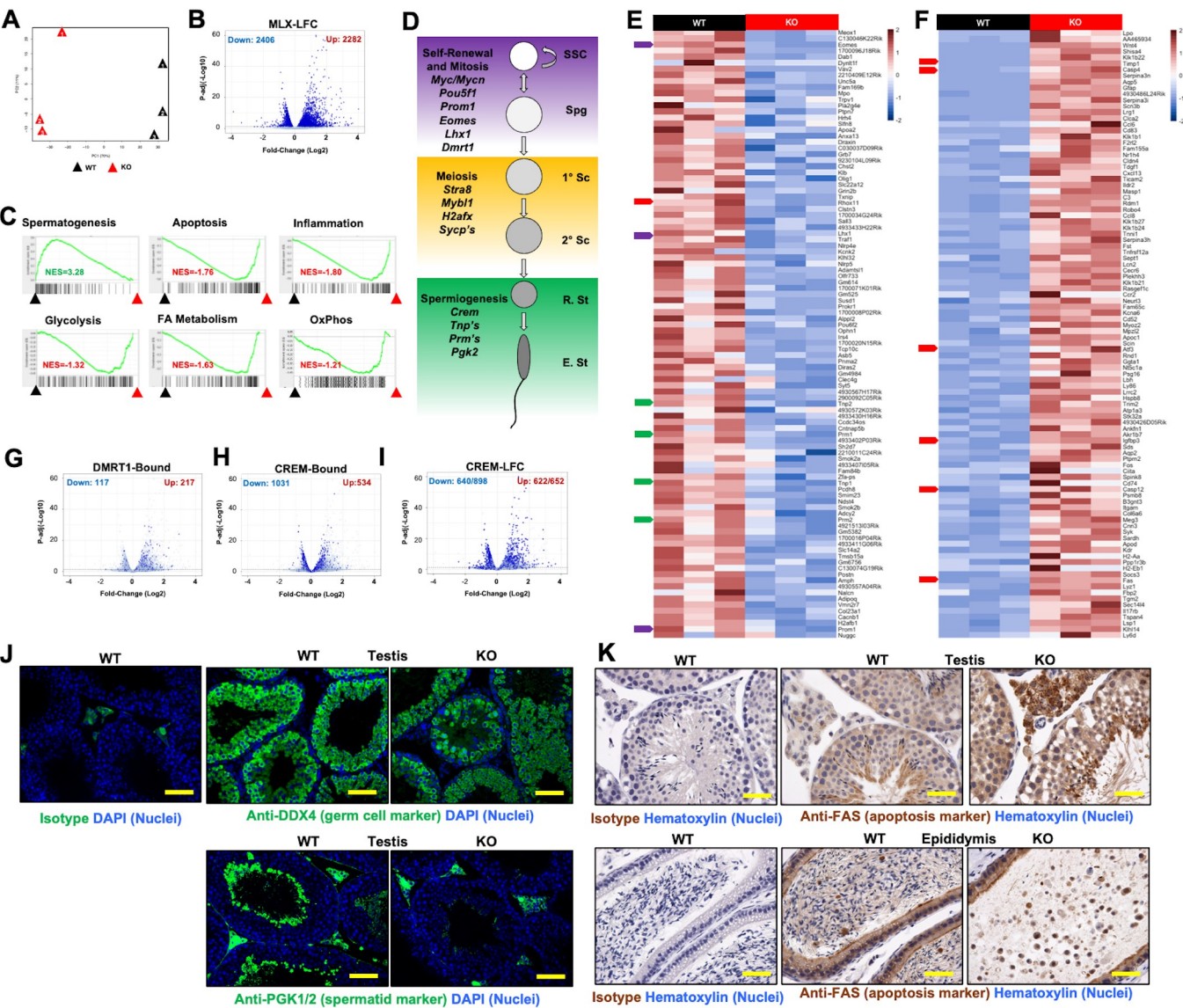

**Fig 5. RNA profiling of testes from WT and MLX^KO mice.** (**A**) PCA of RNA-seq data from WT versus MLX^KO whole testes tissue. (**B**) Volcano plot derived from RNA-seq data showing LFC (x-axis) by −Log10 *p*-value (y-axis) of pair-wise analyzed RNA-seq data. Shown are DEGs significantly up- or down-regulated ($p <= 0.05$). See S2 Table for a complete list. (**C**) GSEA of RNA-seq data with representative up/down categories shown (see S3 Table for a complete list). (**D**) Schematic depicting spermatogenesis broken into 3 major stages: Self-Renewal/Mitosis, Meiosis, and Spermiogenesis with the indicated stage-specific markers shown (cell type abbreviations: E. St, elongating spermatid; R. St, round spermatid; Sc, spermatocyte; Spg, spermatogonia; SSC, spermatogonial stem cell). (**E, F**) Heat maps of 100 most (**E**) down-regulated and (**F**) up-regulated genes in the MLX^KO relative to WT testes. Colored arrows indicated DEGs associated with SSC/Spg function (purple), spermiogenesis (green), and apoptosis (red). (**G, I**) Volcano plots of RNA-seq data as in (**A**) with (shown in dark blue, compared with light blue) DEGs previously reported to be (**G**) DMRT1-bound [30], (**H**) CREM-bound [31], and (**I**) responsive to CREM loss of function CREM-LFC [32]. (**J**) IF analysis of WT and MLX^KO testis stained for the indicated proteins (200×, scale bar = 200 uM). (**K**) IHC analysis of WT versus MLX^KO testis and epididymis stained for the indicated protein (400×, scale bar = 100 uM). The underlying data for Fig 5A–5C and 5G–5I can be found in S1 Data. DEG, differentially expressed gene; GSEA, gene set enrichment analysis; IF, immunofluorescence; IHC, immunohistochemistry; KO, knockout; LFC, log fold change; MLX, MAX-Like protein X; PCA, principal component analysis; RNA-seq, RNA sequencing; WT, wild-type.

changes in the male germ cell tumor (MGCT) cell line NTera2 (Fig 6B), supporting a cell autonomous role for MLX in regulating the expression of these SSC markers. This suggests that MLX may impact stem cell function in male GCs as well as during subsequent differentiation.

We also assessed the expression of metabolic and stress targets identified by RNA-seq, as well as markers of spermatogenesis, in the seminiferous tubules by western blot (WB) analysis of isolated cells from WT and MLX$^{KO}$ mice. As shown in Fig 6C, the known MLX target TXNIP is decreased, and the marker of fatty acid beta-oxidation CPT1A is increased along with stress-related proteins including FAS, BIM, IGFBP3, and γH2AX concomitant with PARP cleavage, all of which are consistent with increased apoptosis. We also confirmed decreased expression of the mature St/spermatozoa marker PGK2, while the pan-GC marker DDX4 is not significantly altered. This further confirms a disruption of normal differentiation associated with elevated stress.

In contrast with our observations in seminiferous tubules, cells isolated from epididymides did not robustly express MYC network or stem cell markers. However, as shown in Fig 6D, MLX$^{KO}$ epididymal cells did exhibit alterations to the same metabolic targets TXNIP and CPT1A, as well as elevated stress markers FAS, BIM, IGFBP3, γH2AX, and PARP cleavage. Epididymal cells from MLX$^{KO}$ mice also maintained the immature GC marker DDX4 (which is normally absent from epididymal cells of WT mice) and, consistent with spermiogenic defects, they also lacked PGK2. Interestingly, MLX appears to regulate many of the same proteins in cells isolated from the interstitium of the testes, supportive of a broad role for MLX in both stabilizing its binding partners and regulating metabolic targets (S5A Fig).

In summary, MLX appears to regulate male GC function at multiple stages: While MYCN-MAX are expressed in the primitive Spg population, loss of MLX destabilizes MYCN, thereby potentially affecting stem cell function. MLX also appears to suppress stress as early as the initiation of meiosis, as γH2AX is induced then, but should resolve after the completion of genome reduction. While the genomes of MLX$^{KO}$ St are indeed reduced to haploid, the stress markers associated with meiosis (and others) are maintained. This correlates with loss of many late St markers and apoptosis. As Sertoli-specific deletion does not result in widespread apoptosis of GCs, this supports a broad role for MLX in facilitating a cell autonomous survival pathway in the male germline.

## MLX regulates glucose and lipid metabolism and suppresses apoptosis

In order to gauge the functional consequences of alterations to metabolism associated with loss of MLX, targeted LC–MS/MS was utilized to monitor glycolytic and beta-oxidation metabolites in isolated seminiferous tubule cells. As shown in Fig 6E, we detected increased intracellular glucose, consistent with the diminished expression of TXNIP, which is known to suppress glucose uptake, while there was no significant change in pyruvate or lactate levels, perhaps due to decreased expression of glycolytic enzymes that are targets of CREM (e.g., LDHA and LDHC). We also detected a significant increase in a number of acyl-carnitine species (the product of CPT1A enzymatic activity) (Fig 6E), while there was no change in acetyl-carnitine (C2-carnitine), consistent with diminished expression of CRAT (another CREM target down in the MLX$^{KO}$ testes) as opposed to the general up-regulation of fatty acid gene set in general, many of which are regulated by CREM (e.g., XBP1 and SREBF1). These changes are consistent with a role for MLX as a transcriptional regulator of metabolism in the seminiferous epithelium predominantly comprised of Spg, Sc, and St. A hypothetical model for putative targets of MLX responsible for these changes is shown in Fig 6F, including the previously reported positive correlation between MondoA-MLX and TXNIP [8] and the inverse correlation between TXNIP and CPT1A [42].

Because *Fas* has been established as a developmental regulator of cell survival during spermatogenesis [43] and cells from MLX$^{KO}$ testes exhibited increased *Fas* mRNA and protein, we asked whether MLX plays a broader and cell autonomous role in regulating the FAS death

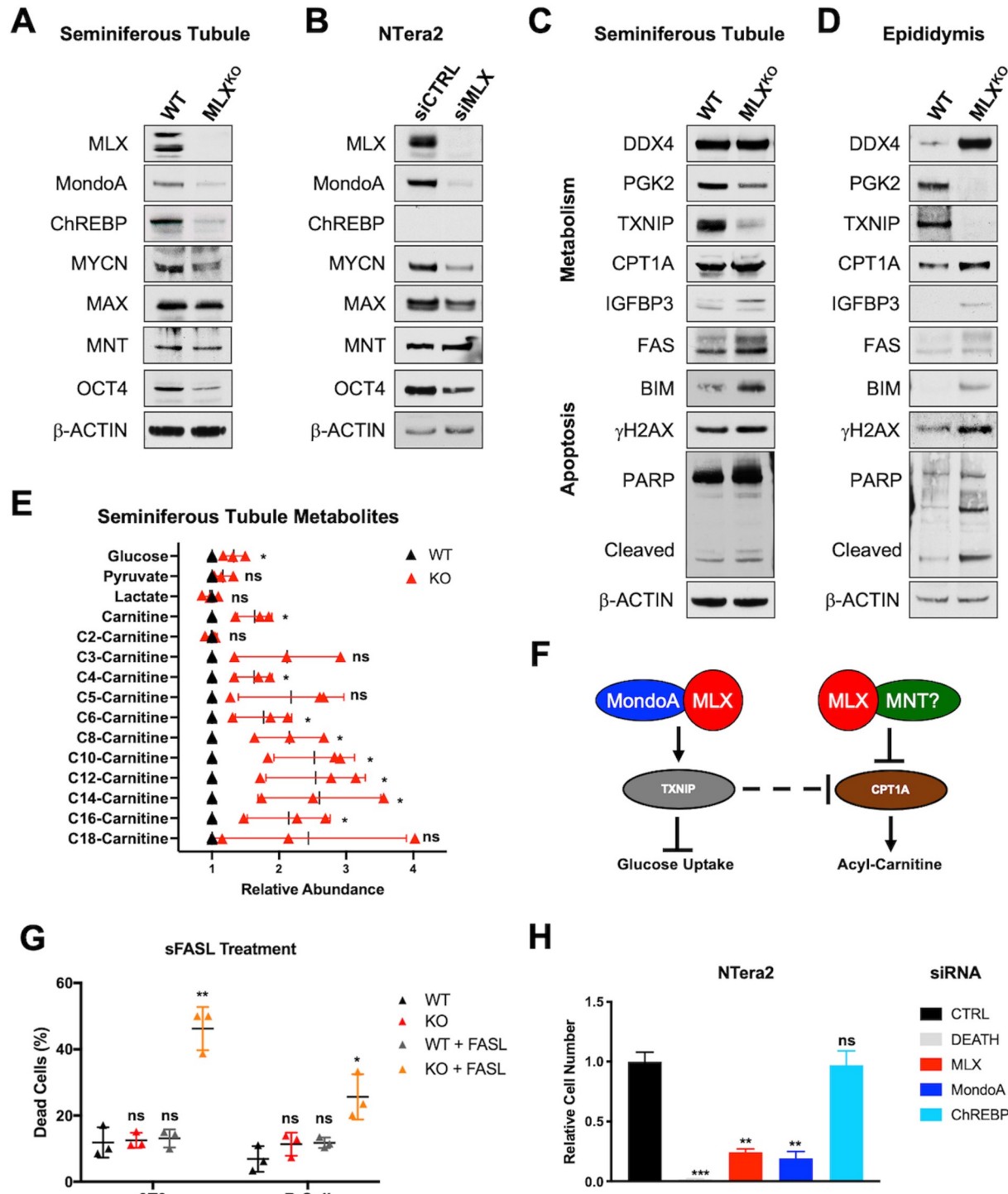

**Fig 6. Analysis of DEGs from WT and MLX^KO mice.** (**A–D**) WB analysis of (**A**) isolated seminiferous tubule cells from WT and MLX^KO mice; (**B**) NTera2 MGCT cells treated with siCTRL or siMLX; (**C**) Seminiferous tubule and (**D**) epididymal cells from WT versus MLX^KO mice probed for the indicated proteins. (**E**) LC–MS relative abundance data for the indicated metabolites from isolated seminiferous tubule cells ($N = 3$ paired littermates, shown is the mean +/− SD, paired $t$ test). (**F**) Model explaining MLX regulation of metabolic targets. (**G**) sFASL treatment of the indicated cell lines ($N = 3$ biological replicates, shown is the percent dead cells with mean +/− SD). (**H**) Relative viable cell number of the NTera2 cells after siRNA transfection with the indicated siRNA. siDEATH included as control for siRNA transfection efficacy ($N = 4$ independent experiments, shown is the mean +/− SEM). One-way ANOVA with a Dunnett correction was used for 6G and 6H (*$p < 0.05$, ** $p < 0.01$, *** $p < 0.001$, *** $p < 0.0001$). The underlying data for Fig 6E, 6G and 6H can be found in S1 Data. DEG, differentially

expressed gene; KO, knockout; LC–MS, liquid chromatography–mass spectrometry; MGCT, male germ cell tumor; MLX, MAX-Like protein X; sFASL, soluble FAS ligand; siRNA, small interfering RNA; WB, western blot; WT, wild-type.

receptor. To this end, we determined the effect of soluble FAS ligand (sFASL) on immortalized 3T3 cells and primary B cells derived from WT versus MLX$^{KO}$ mice. As shown in Fig 6G, FASL selectively kills both MLX$^{KO}$ 3T3 cells and primary B cells while minimally affecting WT cells under normal culture conditions. We note that the MLX$^{KO}$ and WT 3T3 cells are equally viable under standard culture conditions. However, as we previously reported, MLX$^{KO}$ 3T3 cells undergo rapid apoptosis following enforced MYC expression [20]. Importantly, the expression of FAS protein is also elevated in MLX$^{KO}$ 3T3 cells and is suppressed by the reintroduction of any one of the 3 isoforms of MLX into null cells (S6B Fig). Reintroduction of MLX also stabilizes MondoA and MYC-MAX protein levels. Consistent with the established role of MondoA-MLX [20], the MLX target genes TXNIP, TOMM20, and FASN are repressed in the MLX$^{KO}$ cells, but robustly reexpressed with reconstitution of MLX (S6B Fig). This indicates a direct role for MLX in both activation of metabolic targets and suppression of FAS levels and suggests that MLX loss sensitizes cells to context-dependent death, not only as a consequence of MYC activation, but also in response to environmental factors such as FASL and glucose levels. We surmise that MLX normally attenuates stress and apoptosis during spermatogenesis, a process involving high metabolic demand, dependent upon a glycolytic program driven by both MYC and MYCN [39], as well as directly modified by FASL-FAS signaling [43].

## MGCTs require MLX for survival

As male GC progenitors (Spg) are the target of transformation in MGCT development, we extended our observations on the requirement for MLX in spermatogenesis and cellular survival by silencing MLX, and its dimerization partners MondoA or ChREBP, in a MGCT cell line (NTera2). Knockdown of MLX or MondoA in NTera2 cells resulted in a significant reduction in both the expression of SSC markers (Fig 6B, S6 Fig) and in viability (Fig 6H), while siChREBP has no effect on NTera2 cells, as expected, since it is not expressed. These data are consistent with both cell type–specific effects of MondoA, as well as a cell autonomous requirement for MondoA-MLX in GC tumors.

As shown by immunoblot, NTera2 cells treated with siMLX exhibited loss of MLX-dependent metabolic targets, similar to the MLX$^{KO}$ in 3T3 cells (S6B and S6C Fig). Also, as in seminiferous tubule cells in vivo (Fig 6A), the expression of MYCN and MAX decreased upon knockdown of MondoA or MLX in the NTera2 cells (S6C Fig). A broader analysis of MGCTs based on TCGA data shows that *MLX* correlates with *POU5F1*, encoding OCT4, as well as *MYCN* (S6D Fig). Moreover, an independent dataset from oncomine.org [44] also indicates overexpression of *MLX*, *POU5F1*, *MYCN*, *MLXIP* (encoding MondoA), *MAX*, and *MYC* in MGCT (S6E Fig). Immunohistochemistry (IHC) analysis of xenografts derived from NTera2 cells [45] (S6F Fig) shows that all of these factors co-expressed in the pluripotent and proliferative (Ki67+) region of the tumors. This supports a role for MondoA-MLX in viability of GC-derived tumor cells, suggesting that they coordinately regulate proliferation and stemness and suppress cell death.

## MLX directly regulates male GC development in coordination with MAX

To identify genomic binding sites for MLX and MAX, we carried out chromatin immunoprecipitation and sequencing (ChIP-Seq) on WT and MLX$^{KO}$ testes. MLX binding sites were associated with 855 individual gene loci, while MAX peaks were associated with 874 gene loci

in the WT tissue and 627 gene loci in the MLX^KO tissue (association defined as within 5 Kb of the transcription start site (TSS) of a locus) (Fig 7A). Among the subset of gene promoters that exhibited binding by both MAX and MLX (Fig 7A) are promoters of previously reported targets such as *Txnip* and *Arrdc4* as well as loci encoding protamine and transition proteins (*Prm* and *Tnp*, respectively), key genes involved in spermatogenesis (Fig 7B). Differentially expressed transcripts in WT versus MLX^KO testes, as detected by RNA-seq, showed significant correlation with MLX and MAX binding (Fig 7C–7D'), suggesting that these factors contribute directly to the regulation of genes whose expression is altered upon MLX loss. While genes occupied by MAX or MLX in general tend to display decreased expression in MLX deleted testes, many MAX or MLX bound genes were also found to be up-regulated subsequent to MLX loss (Fig 7C and 7D).

Because we had observed a correlation between MLX loss and altered expression of genes regulated by male-specific TFs CREM and DMRT1 (Fig 5G and 5H), we assessed overlap between MLX and CREM and DMRT1 binding sites, revealing highly significant binding by MLX to a subset of CREM-regulated genes (Fig 7E). MAX binding is also apparent at CREM targets, although decreased in the MLX^KO (Fig 7E' and 7G). While the CREM TF binds to a significant number of DMRT1 target genes, few DMRT1 regulated genes are also bound by MLX or MAX (Fig 7E and 7E'). This is consistent with our data indicating that relative to WT, MLX^KO testes exhibit significantly decreased expression of CREM-bound genes but not DMRT1-bound genes (Fig 7F and 7F'). The CREM and DMRT1 TFs are considered to be essential mediators of spermatogenesis and spermiogenesis, and the overlap between MLX and CREM genomic occupancy is consistent with similarities between MLX and CREM loss of function and a proapoptotic phenotype in male GCs [35]. We note that the testes-specific binding to key male GC-specific genes (e.g., *Prm* and *Tnp*) (Fig 7B) is also consistent with the observed male-specific fertility phenotype of MLX null mice, as female GCs do not express protamines or transition proteins known to be associated with sperm-specific genomic compaction.

We also note that in cells from MLX^KO testes, MAX occupancy is altered: MAX binds only 19% (164/874) of its targeted gene loci detected in WT testes (Fig 7A). However, In MLX^KO testes, MAX occupies new sites that it did not occupy in WT testes (Fig 7A). We observed a similar shift in MAX occupancy in MLX^KO 3T3 cells (see below). We find that 60% of genes bound by MAX in MLX^KO testis cells are down-regulated (Fig 7H and 7H'). Of the 53 loci in the MLX^KO testis in which de novo MAX binding appears to "replace" MLX, 9 de novo MAX-bound genes were differentially expressed in MLX^KO that are functionally important in the testes. These include genes known to regulate MYC (*Senp1* [46]) and MYCN (*Ptprd* [47]) and/or OCT4 protein stability (*Senp1* [48], WNT signaling (*Chd11* [49], and *Ctnna3* [50]), promote apoptosis (*Prune2* [51]), and regulate DNA damage response, specifically in spermatogenic cells (*Rnf138* [52,53]) (see S4 Table). These findings indicate that the cellular response to MLX loss involves a shift in the MAX dependent transcriptional machinery to genes involved in stress response.

In order to more broadly delineate the pathways in which MLX and MAX bound genes are involved, we utilized Enrichr analysis [34] of both the Molecular Signature Database (MSigDB) as well as CHEA database of our ChIP-Seq data to identify pathways altered, as well as transcriptional regulators with a high likelihood of associating with MAX and MLX target genes. MLX targets enriched for spermatogenesis, Notch signaling, and mitotic spindle from the MSigDB (S7A Fig), while MAX(WT) targets also enriched for mitotic spindle, as well as Myc Targets V1 (S7B Fig). In contrast to MAX(WT), which overlapped with MLX, MAX(MLX^KO) only enriched for PI3K/AKT/mTOR Signaling (S7C Fig), suggestive of a loss of coordination with MLX and a gaining of altered signal transduction targets. The 53 de novo bound sites did

not enrich for any significant pathways in MSigDB (S7D Fig). However, the same list does enrich for potential transcriptional regulators from the CHEA database. As shown in S7A'– S7D' Fig, while MLX enriches for CREM and SOX9 targets (consistent with a role for MLX in both Sertoli and GCs), MAX(WT) overlaps with MLX somewhat (e.g., PRDM5 targets), MAX (WT) also targets genes shared E2F1. Intriguingly, both PRDM5 and E2F1 enrichment are lost in MAX(MLX$^{KO}$) while gaining binding to TCF4(WNT) and SOX9 targets. Upon analyzing just the 53 de novo targets, they further enrich for other stress pathway regulators important for testes function, such as AR and STAT3 (S7D' Fig). The lists of potentially overlapping TFs shown in S7A–S7D Fig indicate that many of the bound loci are known to be transcriptional targets of stress responsive TFs, which we found to be up-regulated in MLX$^{KO}$ testis in our GSEA analysis. The analysis also indicates that (i) MLX targets spermatogenesis and Sertoli cell pathways; (ii) MAX alone, as well as in combination with MLX, targets essential spermato-genesis functions in WT testis; and (iii) MAX partially shifts to Sertoli and stress pathways upon MLX deletion (see S4 Table).

## MNT and MLX TFs bind metabolic and stress targets shared with MAX

Our data showing that the majority of loci bound by MAX in MLX$^{KO}$ testis cells are down-regulated (Fig 7H and 7H') raise the possibility that loss of MLX promotes heterodimerization of MAX with a transcriptional repressor within the network. To better understand the role of other network members in modulating MLX activity, we extended our genomic occupancy analysis to WT and MLX$^{KO}$ 3T3 cells, a system that permits us to directly assess the effect of manipulating the expression of network members. One such network member is the transcriptional repressor MNT, which has been shown to independently dimerize with MAX as well as with MLX and also to form MNT-MNT homodimers [54]. MLX, MAX, and MNT exhibit a similar genomic distribution of occupancy proximal and downstream of the TSS within regions significantly enriched for the E-Box sequence motif (Fig 8A and 8B). MAX binds to the largest share of loci, while MLX and MNT occupy a subset of loci occupied by MAX (Fig 8C). Moreover, data from individual tracks indicate that multiple MYC network members can occupy the same promoter regions and exhibit subtle changes in occupancy (in the case of the *Txnip* promoter) upon MLX deletion (Fig 8D). Interestingly, as observed for MAX in testes (Fig 7A), the genes bound by both MAX and MNT shift between WT and MLX$^{KO}$ 3T3 cells (S8A Fig). CHEA analysis of the 69 de novo MAX binding sites in MLX$^{KO}$ 3T3 cells implies a functional conservation with de novo MAX pathways in the testis (S7D and S8C Figs). Both MLX and MNT deletions independently induce increased protein expression of BIM and the intrinsic stress response protein ATF4 (Fig 8E), suggesting that MNT–MLX heterodimers contribute to the response to stress. Notably, loss of MNT induces TXNIP (which is lost upon MLX deletion), indicating that it likely functions in concert with MAX and/or MLX to suppress TXNIP expression, thereby promoting glycolysis (see model in Fig 8F). Indeed, in 3T3 cells, MNT and MAX co-occupy other important MLX targets (*Arrdc4*, *Fasn*, and *Atf4*) (S8B Fig). MNT-bound loci in WT 3T3s also enrich for multiple stress-related pathways (S8D Fig). However, in MLX$^{KO}$ cells, MNT binding largely enriches for a different set of pathways, again consistent with the idea that MNT and MLX coordinately regulate the stress response (S8E Fig).

To further explore the potential role of MNT in apoptosis suppression in GCs, we treated male GC-derived lines with siRNAs against MLX and MNT. While no effect was observed in GC-2 mouse Sc-like lines treated with siRNA against MLX, MondoA, or MNT (S9A Fig), siRNA against MNT in the GC-1-spg Spg-like mouse GC line decreased viability to a similar extent as siRNAs against MLX (Fig 9A). Moreover, siMNT treatment of GC-1-spg cells

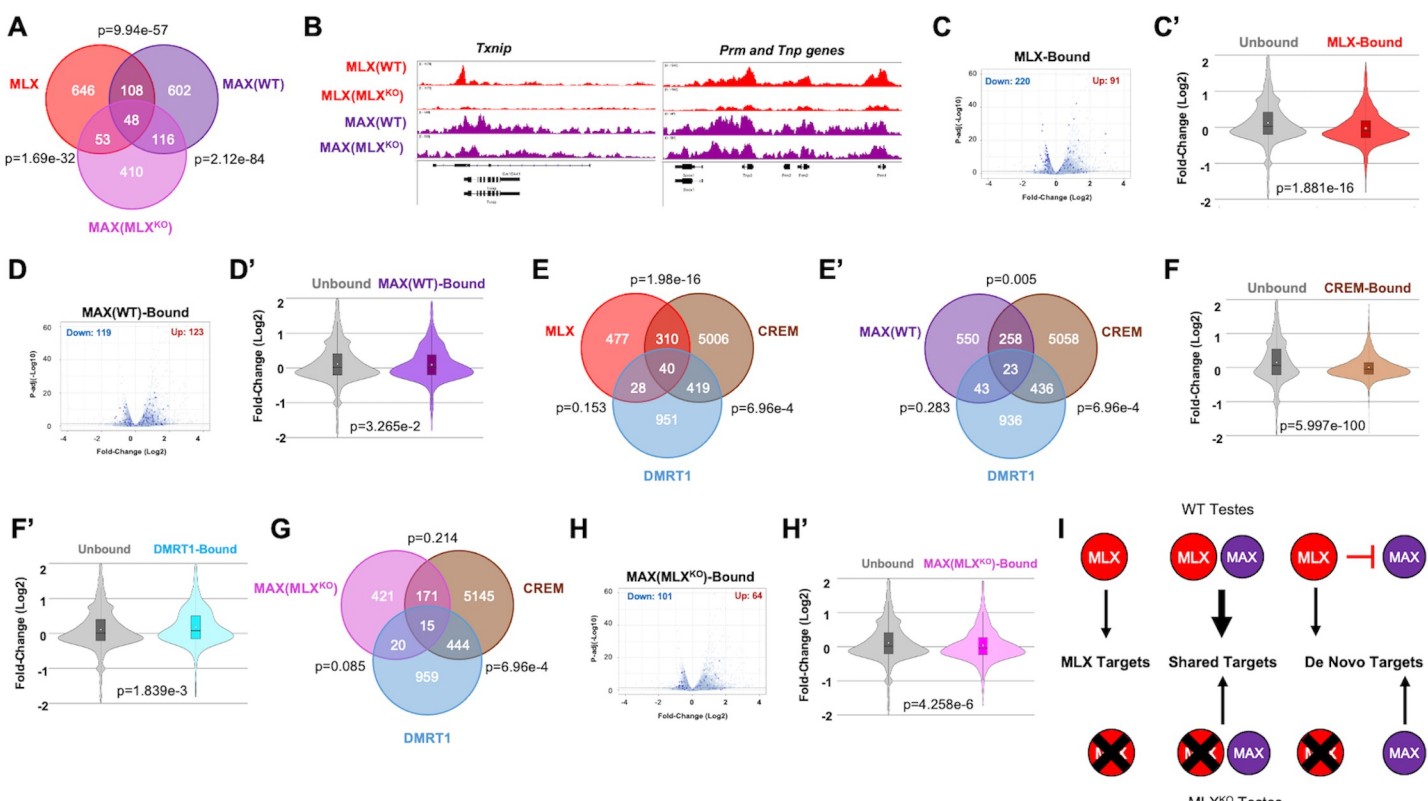

**Fig 7. Analysis of MLX binding and regulation of essential spermatogenesis genes in testes.** (**A**) Venn diagram showing binding overlap of loci occupied by MLX and MAX in WT and MLX$^{KO}$ testes identified by ChIP-Seq. (**B**) Examples of individual ChIP-Seq tracks for *Txnip* and male GC-specific genes (*Prm* and *Tnp*) showing occupancy by MLX and MAX in WT and MLX$^{KO}$ testes. (**C**) Volcano plot showing LFC in mRNA expression between WT versus MLX$^{KO}$ testis with genes that are MLX bound (dark blue dots) compared with those genes not bound by MLX (light blue dots) (KS test p = 0.000000e+00). (**C'**) Violin plots of LFC data from (**C**). (**D**) Volcano plot showing LFC in mRNA expression between WT versus MLX$^{KO}$ testis with genes that are MAX(WT)-Bound in WT testes (dark blue dots) or not bound (light blue dots) (KS test p = 3.853399e-02). (**D'**) Violin plots of LFC data from (**D**). (**E, E'**) Venn diagrams showing (**E**) overlap of MLX bound genes with CREM and DMRT1 bound genes (**E'**) overlap of MAX with known CREM and DMRT1 target genes in WT testes. (**F, F'**) Violin plots comparing LFC of WT versus MLX$^{KO}$ testes for: (**F**) CREM-bound loci and (**F'**) for DMRT1-bound loci (*t* test). (**G**) Venn diagrams showing binding overlap of MAX(MLX$^{KO}$)-bound genes in MLX$^{KO}$ testes with known CREM and DMRT1 target genes in testes. (**H**) Volcano plot showing LFC in mRNA expression between WT versus MLX$^{KO}$ testis with genes that are MAX (MLX$^{KO}$)-bound in MLX$^{KO}$ testis (KS test p = 7.366175e-05). (**H'**) Violin plots comparing LFC of WT versus MLX$^{KO}$ testes for MAX(WT)-bound versus unbound loci. (**I**) Diagram summarizing modes of MLX and MAX binding to subsets of target genes. De novo targets refers to MLX-bound targets only occupied by MAX in MLX$^{KO}$ testis. In WT cells, MLX is predicted to either directly or indirectly exclude MAX from these sites. For Venn diagrams, *p*-values shown are derived from hypergeometric tests. For volcano plots, a KS test was used, and for violin plots, a *t* test was used. The underlying data for Fig 7A–7H can be found in S1 Data. ChIP-Seq, chromatin immunoprecipitation and sequencing; GC, germ cell; KO, knockout; LFC, log fold change; MLX, MAX-Like protein X; WT, wild-type.

strongly induced both TXNIP and BIM (Fig 8C) as we earlier described observed in MNT$^{KO}$ 3T3 cells. These responses may be related to the elevated expression of MNT in GC-1-spg compared with GC-2 cells (S9B Fig). MNT expression, as well as expression of MLX and MondoA, is also required for the survival of human MGCT-derived NTera2 cells (Figs 6H and 9B). Furthermore, loss of either MLX or MNT triggers cell death in the NTera2 cell line by activating similar stress pathways (e.g., BIM, γH2AX, cleaved-PARP) as up-regulated in the tissues and cells derived from MLX$^{KO}$ mice (Figs 8E and 9E). Together, our data support roles for MLX and MNT in the growth and survival of multiple cell types.

We previously reported that MondoA knockdown in MYCN-driven neuroblastoma cells leads to induction of apoptosis due, at least in part, to decreased FASN (fatty acid synthase) expression and attenuated fatty acid biosynthesis. The MondoA deficient neuroblastoma cells were rescued by addition of oleic acid (OA; a monounsaturated C18:1 fatty acid) [20]. We confirmed that this was also the case in MondoA knockdown NTera2 cells (S9D Fig). Because

MLX is a functional heterodimeric partner for MondoA, we next tested whether OA treatment of MLX or MNT knockdown cells affected their growth and survival. As shown in Fig 9D, OA rescued growth of NTera2 cells arrested by knockdown of MLX or MNT. Moreover, OA reversed the decreased levels of MYCN, MAX, OCT4, and FASN expression in siMLX treated NTera2 cells (Fig 9E and 9F). Furthermore, MNT knockdown increased FASN expression (Fig 9G, S9C Fig), while having no effect on MYCN, MAX, or OCT4 levels (Fig 9G). These findings are consistent with TCGA data from MGCTs where higher MLX transcript levels positively correlate with FASN expression, while MNT expression is inversely correlated with FASN (Fig 9H). Taken together, these findings suggest that MNT represses a subset of MondoA-MLX targets (modeled in Fig 9I). Moreover, as fatty acid synthesis is associated with MGCTs in vivo [55], we hypothesize that MLX regulates lipid homeostasis in normal and transformed GCs.

Lastly, to extend our findings on the relevance of network activity to human male fertility, we investigated *MNT* expression in the Human Protein Atlas scRNA-seq database, which showed high GC expression, enriched in the Spg and Sc (S9E Fig). We next examined a dataset comparing normospermic to teratozoospermic men [56] that indicated a significant correlation of male sterility with decreased levels of mRNA encoding MLX, MNT, as well as CREM and its targets (including *TNP* and *PRM* transcripts) and increased levels of BCL2L11, FASN, and XBP1 (S9F Fig). These findings match many of the key changes observed in the MLX$^{KO}$ testes compared with WT and are consistent with a role for MNT–MLX interactions and their shared transcriptional targets in aspects of human male fertility by acting as regulators of mammalian spermatogenesis.

## Discussion

Here, we describe a previously unexplored function of the MLX-based arm of the extended MYC network, namely an absolute requirement for MLX (and MondoA) in normal testis development and function. Inactivation of the MLX arm of the network leads to male-specific sterility, altered metabolism, and increased stress, accompanied by widespread activation of apoptosis in spermatogenesis. MYC, MYCN, MGA, and MAX have all been shown to play distinct and critical roles in male GC development. MYC/MYCN (and MAX) regulate SSC function, metabolism, and proliferation through PDPK1 [39], whereas MAX (heterodimerized with MGA) represses meiosis-associated transcription [57]. All 4 of these genes are also essential for normal embryonic development [16,17,58–60]. By contrast, MLX (this report) and MondoA [19] are both dispensable for embryogenesis (Fig 1). Our characterization of spermatogenesis in MLX$^{KO}$ testes, as well as our genetic and genomic analyses, suggests that infertility is due to accumulation of cell autonomous defects in developing sperm as well as to a loss of Sertoli cell function in supporting survival and differentiation at multiple stages of spermatogenesis. The model depicted in Fig 10 posits that MLX loss in Spg and primary Scs leads to an inability to properly complete meiosis and a failure to produce normally differentiated round and elongated St and mature spermatozoa.

Human idiopathic OAT, which shares several characteristics with the MLX$^{KO}$ phenotype, is typically associated with a variety of metabolism-related pathologies, including metabolic syndrome, diabetes, obesity, and inflammation [61], all of which have been reported to be linked with dysregulation of MLX's dimerization partners MondoA and/or ChREBP (reviewed in [62,63]). While *Mlx* is not essential for normal embryonic development, previous studies have shown that MLX loss does lead to markedly diminished viability in a subset of cell types in vivo and ex vivo in the context of intrinsic and extrinsic metabolic stress (e.g., splenocytes, 3T3 cells, and tumors (e.g., MYCN-amplified neuroblastomas and MGCT)). This is consistent with

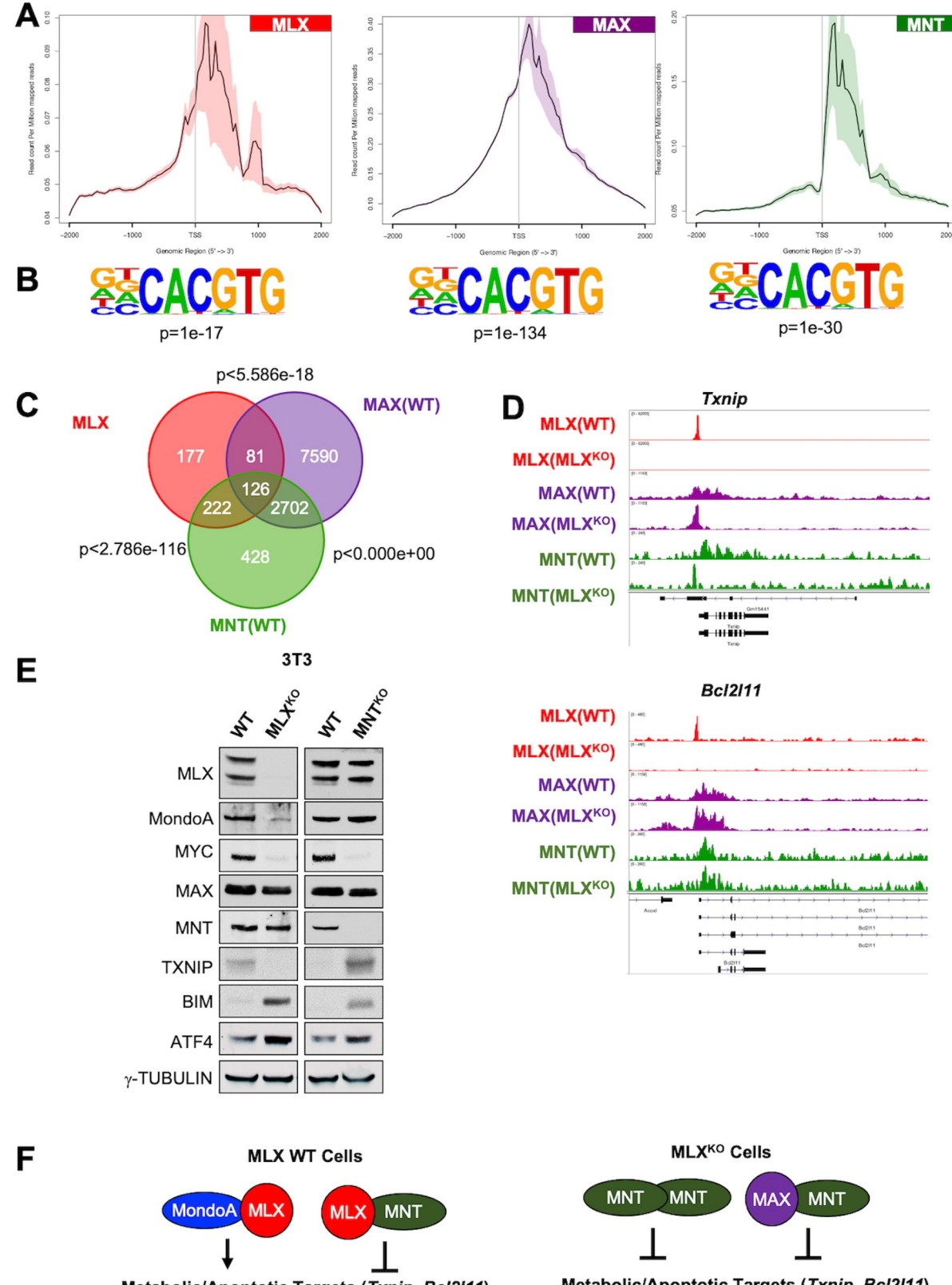

**Fig 8. MLX shares transcriptional targets with MAX and MNT.** (**A**) Meta-plots of TSS occupancy for the indicated TFs in WT 3T3 cells. (**B**) E-Box Motif derived from HOMER analysis with the indicated *p*-value. (**C**) Venn diagram indicating overlap of genes bound by MLX,

MAX, and MNT in 3T3 cells with *p*-value calculated from a hypergeometric test. (**D**) ChIP-Seq tracks for MLX, MAX, and MNT on the indicated gene promoters from WT and MLX$^{KO}$ 3T3 cells. (**E**) WBs from WT and MLX$^{KO}$ and WT and MNT$^{KO}$ littermate 3T3 cell lines probed for the indicated proteins. (**F**) Diagram summarizing modes of MondoA, MNT, and MLX association and consequent transcriptional responses. The underlying data for Fig 8A–8D can be found in S1 Data. ChIP-Seq, chromatin immunoprecipitation and sequencing; E-box, Enhancer box; KO, knockout; MLX, MAX-Like protein X; TF, transcription factor; TSS, transcription start site; WB, western blot; WT, wild-type.

the key role of nutrient sensing in male spermatogenesis and suggests possible routes of dysregulation associated with idiopathic OAT in humans.

Relevant to a critical role for MLX in metabolism during spermatogenesis, we observed that the serum and testes of MLX$^{KO}$ mice exhibit altered abundance of metabolites, consistent with a shift in glucose metabolism from oxidative phosphorylation, via the tricarboxylic acid (TCA) cycle, to the production of lactate and to oxidation of alternative TCA substrates, such as branched-chain amino acids and fatty acids. Deletion of MondoA in mice leads to a similar change in serum metabolites, associated with enhanced glycolysis and activation of beta-oxidation [10,19]. Moreover, loss of the MondoA-MLX target TXNIP is sufficient to induce a similar metabolic profile, including augmented CPT1A expression and activity [42]. This suggests that the choice of oxidative substrate for the TCA cycle in testis is controlled at least in part by MondoA-MLX through its regulation of TXNIP. These metabolic alterations, observed in the MLX$^{KO}$ testes, are associated with GC apoptosis, concomitant with aseptic inflammation and immune cell activation, symptoms also associated with male infertility (reviewed in [64]). While MLX and TXNIP have been linked to inflammation [15], loss of TXNIP alone does not result in male sterility [65]. This suggests that critical MLX transcriptional targets in addition to TXNIP are responsible for the majority of the male-specific sterility phenotypes associated with loss of either MLX or MondoA. An example of such is *Arrdc4*, a bona fide MondoA-MLX target, bound by MLX in our dataset, and it has recently been shown to be required for normal male fertility in the mouse [29].

Our earlier work in MYCN amplified neuroblastomas following MondoA or MLX knockdown linked growth arrest and apoptosis with attenuated lipid biosynthesis that could be rescued by OA. Here, we find that MLX$^{KO}$ testes produce significantly increased levels of multiple acyl-carnitine species and increased levels of carnitine palmitoyltransferase (CPT1A) relative to WT, indicative of altered lipid metabolism and increased stress. Indeed, many DEGs directly bound by MLX have been shown to be involved in lipid metabolism and in spermatogenesis (see S4 Table and references therein). Therefore, altered expression of MLX-dependent target genes controlling lipid metabolism may be at least 1 contributor to the stress accompanying MLX loss of function. Moreover, our data showing that MGCTs such as NTera2 are dependent on MLX, and whose growth arrest upon MLX knockdown is reversible by OA treatment, underscore the importance of lipid synthesis in MLX dependence.

Our transcriptional profiling and genomic occupancy analyses using ChIP-Seq has identified many other direct targets (i.e., genes bound and regulated) of MLX in the testes, including metabolic and stress effectors as well as many genes relevant to male-specific GC development (see S4 Table). Among the latter are a small subset of approximately 5,800 genes previously shown to be bound by the essential transcriptional regulator of spermiogenesis, CREM [31]. These MLX/CREM shared targets include genes encoding factors such as protamines and transition proteins as well as enzymes critical for apoptosis, mitochondrial activity, glucose, and lipid metabolism. All of these have been shown to be present and functional in spermatogenesis (S4 Table and references therein). The MYC/MYCN and CREM target PDPK1 is similarly bound by MLX and MAX and down-regulated in the MLX$^{KO}$ testes (S4 Table). PDPK1 is required for SSC function and also stabilizes MYC protein [39,66]. Important for metabolic

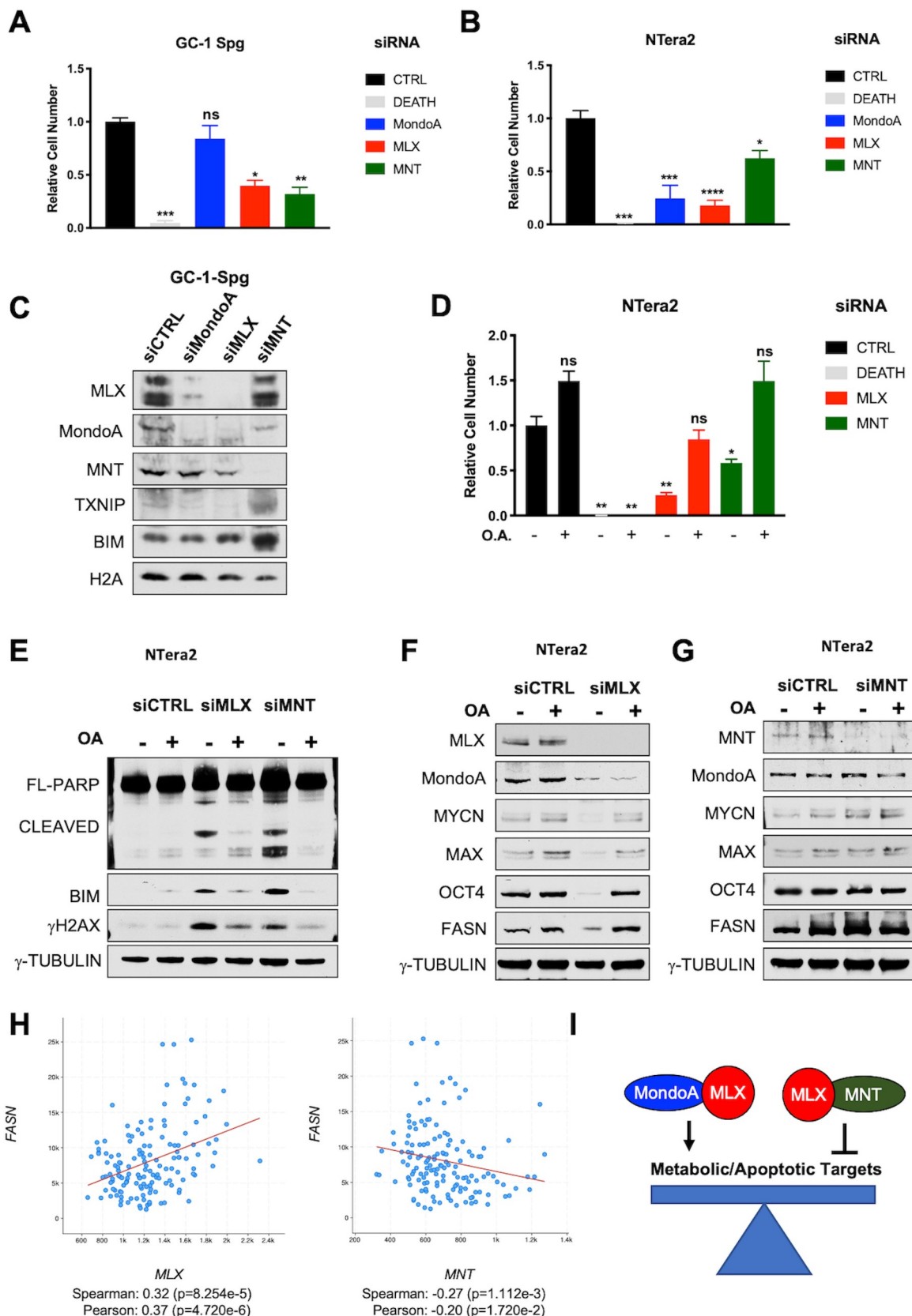

**Fig 9. MLX and MNT regulate metabolism and survival of male GC lines.** (**A**) Relative viable cell number of the GC-1-Spg cells after siRNA transfection with the indicated siRNA (*N* = 3 independent experiments, shown is the mean +/− SD). (**B**) Relative viable NTera2 cell numbers after siRNA transfection with the indicated (*N* = 4 independent experiments, shown is the mean +/− SEM). (**C**) WB analysis of GC-1-Spg cells transfected as in (**A**) probed for the indicated proteins. (**D**) Relative viable cell number of the NTera2 cells after siRNA transfection with the indicated siRNAs. Cells were cultured in the presence or absence of 35 uM OA (*N* = 4 independent experiments, shown is the mean +/− SEM). (**E–G**) WB analysis of NTera2 cells as in (**D**) probed for the indicated proteins. (**H**) Co-expression analysis of MLX and FASN and MNT and FASN from the Testicular Germ Cell Tumor Dataset (TCGA PanCancer Atlas). (**I**) Model depicting modes of MLX activation versus repression and the role of MondoA and MNT in these processes. For all, *p*-values shown from 1-way ANOVA with a Dunnett test compared with the Control (* $p < 0.05$, ** $p < 0.01$, *** $p < 0.001$, *** $p < 0.0001$). The underlying data for Fig 9A, 9B, 9D and 9H can be found in S1 Data. GC, germ cell; OA, oleic acid; siRNA, small interfering RNA; Spg, spermatogonia; WB, western blot.

regulation of sperm motility are a number of sperm-specific glycolytic enzymes, such as PGK2 and GAPDHS. MLX targets *Gapdhs*, and loss of GAPDHS results in infertility and nonmotile spermatozoa [67]. Intriguingly, loss of either PGK2 or GAPDHS results in decreased motility, as well as elevated acyl-carnitine levels [68], similar to MLX^KO. This is supportive of multiple levels of cross talk between glucose, glucose sensing, and lipid metabolism. MLX also shares target genes with CREM including a number of phospholipases required for male fertility including *Ddhd1* [69] and *Plcb1* [70]. These data strongly support the notion that MLX and its binding partners act as transcriptional mediators of critical events in mammalian spermatogenesis in concert with other transcriptional regulators such as CREM and MAX. Interestingly, neither MLX nor MAX occupy a significant number of loci that are targets of the DMRT1 testes-specific transcriptional regulator of differentiation. Thus, the direct activity of MLX and other MYC network members in spermatogenesis may, to some extent, be more focused on functions regulated through the CREM pathway.

Many of the MLX-linked metabolic and stress targets observed in testes are not cell type specific, and we find their expression altered in both seminiferous tubules and testicular interstitial cells, as well as in 3T3 cells derived from WT versus MLX^KO embryos. MLX loss of function sensitizes these cells to apoptotic stimuli and effectors known to be dependent upon MYC (e.g., FASL–FAS interactions and BIM) [71,72]. Taken together, these data support not only a direct role for MLX in regulation of metabolic targets but also a role for transcriptional repression of apoptosis effectors (such as BIM), most likely mediated by MLX heterodimerization with repressors such as MNT.

Our genome-wide occupancy analyses demonstrate that genes bound by MLX are also bound by other members of the MYC network, such as MAX and MNT. Intriguingly, in the absence of MLX, we detect a shift in the occupancies of these other factors: For example, in MLX^KO 3T3 cells, MAX and MNT bind to loci that they did not occupy in WT cells. Our analysis of loci newly occupied by MAX (de novo sites) following MLX deletion show that these comprise many genes implicated in the response to stress (S4 Table, blue highlighted section), suggesting a modulation in transcriptional programming upon MLX loss. Of particular note are loss of *Rnf138* and induction of *Prune2* whose altered expression would be expected to promote spermatogonial apoptosis [51,52]; *Cdh11*, a proapoptotic inhibitor of catenin signaling [49]; and *Acadl*, induced along with other beta-oxidation enzymes in Sertoli cells by phagocytosis of dead GCs [73]. Importantly, this phagocytosis by Sertoli cells is required for male fertility to recycle lipid and other metabolites to the developing primitive GCs, as blocking it results in infertility [68].

Our study further implicates the extended MYC network, and specifically its nutrient-sensing MLX arm, in the direct regulation of, and linkage between, differentiation, metabolism, and apoptosis. Importantly, metabolic programs change along with changes in cellular state and thus must be responsive to both extrinsic signals, such as mitogenic cues (via effectors

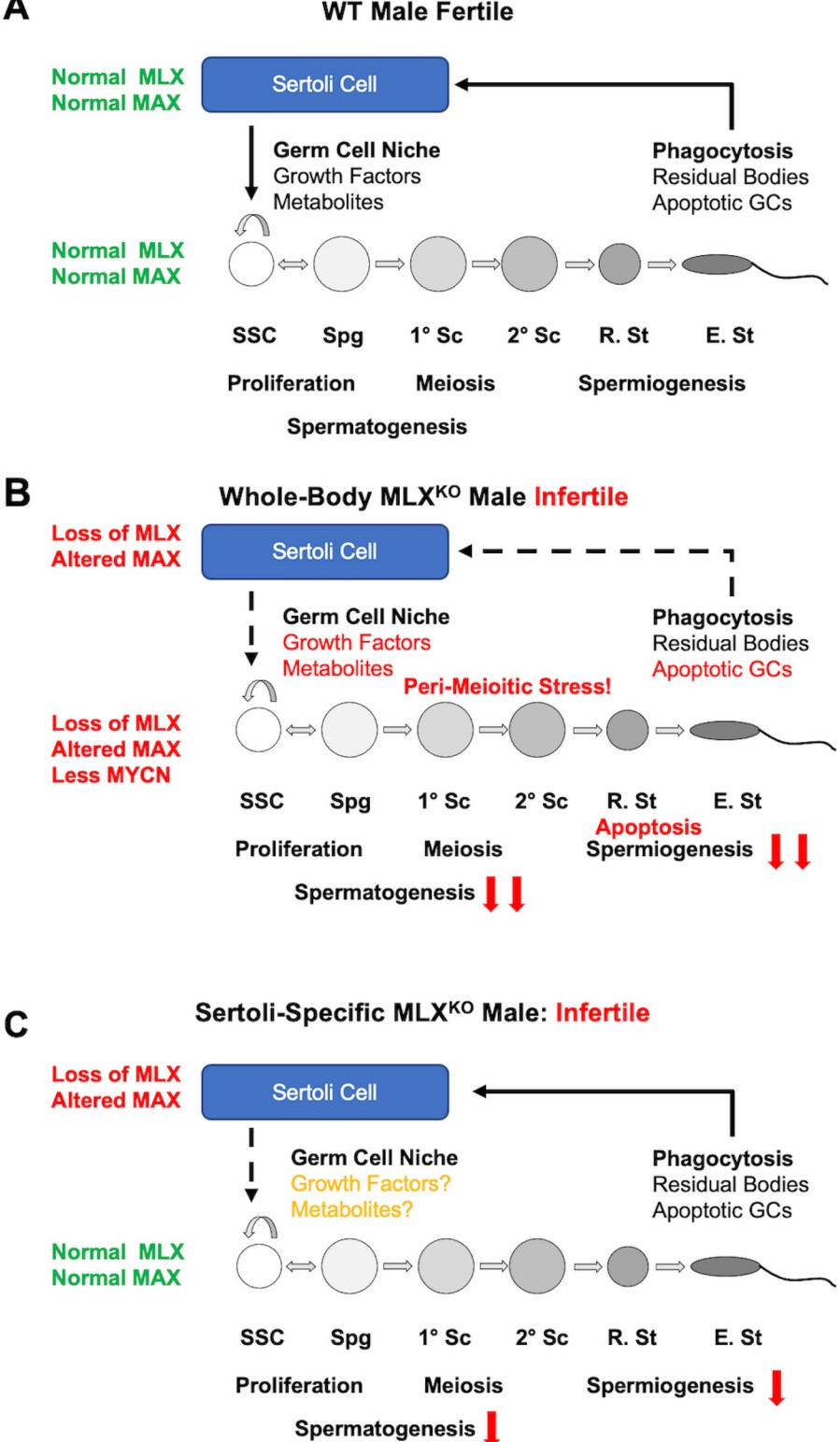

**Fig 10. Models depicting the role of MLX in Sertoli and GC interactions during spermatogenesis. (A)** In normal WT male mice, MLX and MAX collaborate in both the Sertoli cell and spermatogenic cells to maintain the stem cell (SSC) pool as well as facilitate proper differentiation of Spg to Sc and subsequent R. St and E. St. Sertoli cells provide both the spermatogenic niche, as well as the recycling of critical components through phagocytosis of germ cell–

derived residual bodies and apoptotic cells. These features are critical for male fertility, and there is a tight coordination between the Sertoli and GCs at every stage of development. (**B**) Constitutive deletion of *Mlx* leads to infertile male MLX[KO] mice. This is driven by loss of MLX targets, decreased MYCN expression, alterations to MAX function leading to perturbation of normal metabolism, GC apoptosis, and lack of proper spermiogenesis and maturation. There is also shedding of immature cells from the testis to the epididymis, suggestive of altered phagocytosis by Sertoli cells. Sertoli dysfunction is supported by the fact that, as shown in (**C**), Sertoli-specific *Mlx* deletion also results in male infertility. However, the block to DSP and the extent of apoptosis are significantly less severe than observed in the constitutive deletion (B) underscoring the cell autonomous requirement for MLX in GC differentiation. DSP, daily sperm production; E. St, elongating spermatid; GC, germ cell; KO, knockout; MLX, MAX-Like protein X; R. St, round spermatid; SSC, spermatogonial stem cell; WT, wild-type.

such as MYC-MAX), but also to intrinsic metabolic cues (as in the case of G6P and acidosis, known to activate MondoA-MLX [74]), or formation of lipid droplets (demonstrated to sequester and inhibit activity off MondoA-MLX) [75]. It is clear that functional interactions among MYC network members are relevant, not only in MYC-driven oncogenesis, but also during testes development (this work), and regeneration of both skeletal muscle and liver [76,77]). We anticipate that further genetic perturbation of the network in distinct biological contexts, coupled with high-resolution genomic analysis, will yield important insights into the molecular control of normal and abnormal cellular behavior in both tissue homeostasis and oncogenesis.

## Materials and methods

### Animal use

All experiments involving mice were carried out under accordance with the guidelines of the following institutions: Ethics statement: This study was performed in strict accordance with the recommendations in the Guide for the Care and Use of Laboratory Animals of the National Institutes of Health. All of the animals were handled according to approved institutional animal care and use committee (IACUC) protocol 50783 of the Fred Hutchinson Cancer Research Center. The animal experiments conducted at the University of Utah were performed under IACUC protocol 15–04012. Mice were euthanized by asphyxiation under carbon dioxide according to IACUC protocol. Every effort was made to minimize pain and suffering.

### Generation of *Mlx*$^{-/-}$ mice

The *Mlx* KO allele was created in our lab by a homologous recombination targeted method in 129S4 AK7 murine ESCs. The targeting construct is in the PGKneoF2L2DTA backbone and is based upon the coding sequence of *Mlx* transcript variant 1 (encoding MLX-α protein). This construct includes in 5′ to 3′ order: (1) a 1,546 base pair (bp) 5′-homology arm including exons 1 and 2 of *Mlx*; (2) a 1,717 bp Loxp-flanked region encoding exons 3 to 6 (bHLHLZ domain) of *Mlx*; (**3**) a FRT-flanked PGKNEO-positive selection cassette; and (4) a 2,873 bp 3′ homology arm spanning exon 7 of *Mlx* transcript variant 1 and a pGKDTA-negative selection cassette. Selected ESC clones were injected into blastocysts to generate chimeric animals. These chimeras were bred to ROSA26 FlpO/FlpO females [78] in the 129S4 co-isogenic background to remove the Frt-flanked NEO cassette to generate conditional KO mice or Meo2-Cre delete mice [79] to generate total KO mice. The FlpO allele or the Meox2-Cre allele was subsequently crossed out. Two independent mouse lines from independent ESC clones were found to be phenotypically indistinguishable. All the targeting mice were confirmed by PCR and Southern blotting with 5′ external, 3′ external, and NEO probes. To generate Sertoli cell–specific deletion of *Mlx*, floxed animals were bred with mice expression transgenic Cre-recombinase under the control of the *Amh* promoter [26].

## Genotyping

The following primers were used as a mixture of 3 primers for PCR to discriminate WT from KO from floxed animals; MLX forward (MLX 5644f): 5′ actccaggaaaagtgtagctgcc 3′, MLX reverse (MLX 5845r): 5′ caagctgttggcttccatacagg 3′, MLX deletion (MLX 3679f): 5′ caaccatggt-cacacctggttc 3′ yielding the following size PCR products: MLX forward + MLX reverse: WT = 201 bp, flox = 327 bp MLX deletion + MLX reverse: KO = 589 bp. The annealing temperature used was 65°C.

## Mating tests

Pairs of sexually mature mice (1 of each sex) were housed in the same cage, and mating was observed with the seminal plugs and number of pups born recorded. For mendelian frequency, heterozygotes were bred with heterozygotes, and the progeny (F1) were genotyped. For KO mating tests, homozygous null males or females were bred with WT mice.

## Metabolomics

**Reagents.** Acetonitrile, ammonium acetate, and acetic acid (LC–MS grade) were all purchased from Fisher Scientific (Pittsburgh, Pennsylvania, United States of America). The standard compounds corresponding to the measured metabolites were purchased from Sigma-Aldrich (St. Louis, Missouri, USA) and Fisher Scientific.

**Serum sample preparation and LC–MS/MS measurement.** Male mice (3 of each genotype WT, MLX$^{KO}$) were bled by retro-orbital eye bleed, and serum was isolated, then flash-frozen on dry ice before subsequent use for LC–MS/MS. Frozen serum samples were first thawed overnight at 4°C, and 50 μL of each sample was placed in a 2-mL Eppendorf vial. The initial step for protein precipitation and metabolite extraction was performed by adding 500 μL MeOH and 50 μL internal standard solution (containing 1,810.5 μM $^{13}C_3$-lactate and 142 μM $^{13}C_5$-glutamic acid). The mixture was then vortexed for 10 seconds and stored at −20°C for 30 minutes, followed by centrifugation at 14,000 RPM for 10 minutes at 4°C. The supernatants (450 μL) were was collected into a new Eppendorf vial and dried using a CentriVap Concentrator (Labconco, Fort Scott, Kansas, USA). The dried samples were reconstituted in 150 μL of 40% PBS/60% ACN. A pooled sample, which was a mixture of all serum samples, was used as the quality control (QC) sample.

The LC–MS/MS experimental procedures were well documented in our previous studies [20,80–85]. Briefly, all LC–MS/MS experiments were performed on a Waters Aquity I-Class UPLC-XenoTQ-S micro (Waters, Milford, Massachusetts, USA) system. Each sample was injected twice, 10 μL for analysis using negative ionization mode and 2 μL for analysis using positive ionization mode. Both chromatographic separations were performed in hydrophilic interaction chromatography (HILIC) mode on a Waters XBridge BEH Amide column (150 × 2.1 mm, 2.5 μm particle size, Waters). The flow rate was 0.300 mL/min, auto-sampler temperature was kept at 4°C, and the column compartment was set at 40°C. The mobile phase was composed of Solvents A (5 mM ammonium acetate in 90%$H_2O$/ 10% acetonitrile + 0.2% acetic acid) and B (5 mM ammonium acetate in 90%acetonitrile/ 10% $H_2O$ + 0.2% acetic acid). After the initial 2 minutes isocratic elution of 90% B, the percentage of Solvent B was linearly decreased to 50% at t = 5 minutes. The composition of Solvent B maintained at 50% for 4 minutes (t = 9 minutes), and then the percentage of B was gradually raised back to 90%, to prepare for the next injection. The mass spectrometer is equipped with an electrospray ionization (ESI) source. Targeted data acquisition was performed in multiple reaction monitoring (MRM) mode. We monitored 121 and 80 MRM transitions in negative and positive mode, respectively (201 transitions in total). The whole LC–MS system was controlled by MassLynx

software (Waters). The extracted MRM peaks were integrated using TargetLynx software (Waters).

**Cell sample preparation and LC–MS/MS measurement.** Isolated cells from the seminiferous tubules of age-matched WT and MLX$^{KO}$ mice ($N$ = 3) were separated into 4 technical replicates of $1 \times 10^6$ cells each and flash frozen on dry ice. Soluble metabolites were extracted into 1 ml of 20:80% water:methanol before clearing the insoluble and drying down on a SpeedVac before subsequent LC–MS/MS.

The LC–MS/MS experiments were performed on an Agilent 1260 LC-6410 QQQ-MS (Agilent Technologies, Santa Clara, California, USA) system. Moreover, 5 μL of each sample was injected for analysis using positive ionization mode. Chromatographic separation was performed using a Waters XSelect HSS T3 column (2.5 μm, 2.1 × 150 mm). The flow rate was 0.3 mL/min. The mobile phase was composed of Solvents A (100% H$_2$O with 0.2% formic acid) and B (100% ACN with 0.2% formic acid). After the initial 0.5 minutes isocratic elution of 100% A, the percentage of Solvent A was linearly decreased to 5% at t = 10 minutes. Then the percentage of A remained the same (5%) for 5 minutes (t = 15 minutes). The metabolite identities were confirmed by spiking with mixtures of standard compounds. The extracted MRM peaks were integrated using Agilent Masshunter Workstation software (Agilent Technologies).

**Serum T quantification.** Blood was collected by either retro-orbital eye bleed or cardiac stick and heparinized plasma was isolated. T levels were determined by ELISA at the University of Virginia Center for Research in Reproduction (NICHD Grant # U54-HD028934) Charlottesville, Virginia.

**Cell isolation from testicular tissue.** Testes and epididymides were dissected, defatted, and processed as follows for cellular isolation: For seminiferous tubule cell isolation, testes were dissected, decapsulated, and subjected to an enzymatic digestion to isolate seminiferous tubules from the interstitial stromal cells (including Ledig and immune cells). Isolated tubules were then digested to release a single cell suspension of the seminiferous epithelium including GCs and Sertoli cells. For epididymal GCs, epididymides were dissected, and the caudal portion was cut to release cellular content of sperm. Single cell suspension was filtered to remove tissue yielding predominately mature spermatozoa from WT tissue.

**Spermatogenesis analysis.** Testes and epididymides were dissected, defatted, and processed for analysis as previously described [86]. For calculation of testicular DSP rate (St count), testes were weighed and homogenized in 0.1 M sodium phosphate buffer (pH 7.4) with 0.1% Triton X-100 via 8 strokes of a 15-ml Kontes homogenizer. Homogenization-resistant St were counted on a hemocytometer, and St per gram per testis were calculated [87].

**Cell culture.** 3T3 cell lines, Ntera2, HepG2, GC-1-Spg, and GC-2-Spd(ts) cells were all cultured in DMEM with 10% FBS and penn/strep. Primary B220+ sorted splenic B cells were cultured in RPMI with 15% FBS, penn/strep, beta-mercaptol, and LPS (1 ug/ml from Sigma-Aldrich).

**RNAi transfection.** Flexitube Gene Solutions siRNA mixtures (Qiagen, Germantown, MD, USA) were utilized to knockdown the indicated target. RNAiMax (Qiagen) was used according to the manufacturer's conditions. Cells were counted 72 to 96 hours posttransfection, and viability was monitored by trypan blue exclusion. For OA treatment, Oleic Acid Water Soluble (Sigma-Aldrich) was resuspended in sterile water.

**B220+ cell purification.** Spleens were smashed, and B220+ B cells were purified from the splenocytes using AutoMACS system according to the manufacturer's recommendations. Purity was routinely over 95% pure, as assessed by flow cytometry.

**Flow cytometry.** Testis tissue was prepared as in [88]. Isolated seminiferous tubule or epididymal cells were resuspended in staining buffer with HOECHST, and samples were run on a Canto-2 Flow Cytometer and analyzed by FACS-Diva Software.

**IHC tissue staining.** Testicular and epididymal tissue was fixed in Modified Davidson's Fluid (MDF) as described [89], then embedded in paraffin and sectioned onto slides at 5 micron thickness. Slides were deparaffiinized, rehydrated, and antigen retrieval was utilized. For immunofluorescent IHC, Dako reagents were used (Block, Primary Dilution Buffer, Anti-fade mounting media), and Alexa-Flour 488nm secondary antibodies were used in combination with DAPI for staining. For tissue staining and IHC, samples prepared as described above were submitted to the FHCRC Experimental Histopathology Core and either stained with hematoxylin and eison, or stained with the indicated antibodies, visualized with the cromophore DAB and counterstained with hematoxylin to mark nuclei.

The staining of previously published Xenograft tissue of the NTera2 cells [45] were carried out by the Institute of Pathology of the University Medical Center Göttingen. Briefly, 4 micron thick sections were mounted on slides, deparaffinnized, rehydrated, and antigen retrieval was utilized. The slides were stained with primary antibody then biotinylated secondary antibodies using a REAL Detection System (LSAB+ kit; Dako). The signals were visualized using a REAL Streptavidin Alkaline Phosphatase kit (Dako), while Ki-67 staining was visualized with DAB. All samples were counterstained with hematoxylin, mounted in super mount medium, and analyzed via light microscopy.

**WBs.** Tissues and cell pellets were lysed in RIPA buffer. For whole testes sample preps, tissue was homogenized mechanically to facilitate lysis. Lysates were quantified by BCA assay (Pierce Biotechnology, Waltham, MA, USA) or normalized to cell number for equal loading. Samples were resolved on NuPAGE 4–12% Bis-Tris gradient gel before transferring to Nitrocellulose (0.2 micron). Blots were blocked with 5% Milk in TBST, washed with TBST, and probed with primary and secondary antibody in 5% Milk in TBST. The secondary antibody was HRP conjugated, and chemiluminescent detection was employed. Blots were exposed to Pro-Signal Blotting Film (Genesee Scientific, San Diego, CA, USA).

**RNA extraction and sequencing.** RNA was extracted with Trizol reagent, quantified on a TapeStation. A total of 500 ng of RNA was submitted for library preparation through FHCRC Genomics Core. Libraries were aligned to mm10 using TopHat then processed with EdgeR or DE-Seq. Data were analyzed with GSEA, and volcano and violin plots were generated using ggplot.

**ChIP-Seq.** We performed ChIP-seq as previously described [90] with modifications to improve solubility of TFs, which tended to vary depending on cell type. The chromatin preparations from the testes were from a pool of testes from 6 WT/KO animals, and this material was not treated with MNase. The chromatin preparations from the B cells and 3T3 cell lines were treated with MNase. Briefly, after formaldehyde cross-linking, cell lysis, and chromatin fragmentation with MNase, the final SDS concentration after dilution of total chromatin was increased to 0.25% with addition of 20% SDS stock solution. Sonication was performed in a Covaris M220 focused ultrasonicator for 12 minutes with the following settings: 10% duty cycle, 75W peak incident power, 200 cycles/burst, and 6to 7°C bath temperature. The SDS concentration of the sonicated chromatin solution was readjusted to 0.1% with dilution buffer. Immunoprecipitation was performed on the clarified chromatin (input) fraction from $10 \times 10^6$ cellular equivalents, and DNA was purified was using standard phenol:chloroform extraction. The following antibodies were used for ChIP: MAX C-17/sc-197 (Santa Cruz Biotechnology (Paso Robles, CA, USA), Cat. No. sc-8011X), MLX D8G6W (Cell Signaling Technology (Danvers, MA, USA), 85570S), MNT (Bethyl Laboratories (Montgomery, TX, USA), A303-627A), and the negative control GFP (Cell Signaling Technology, 2956S). We used 10 μg of antibody for each immunoprecipitation. To purified ChIP DNA, we added 10 pg of spike-in DNA purified from MNase-digested chromatin from *Drosophila melanogaster* S2 cells or *Saccharomyces cerevisiae* [91] to permit comparison between samples. DNA was then subjected to

library preparation as previously described [92,93], and 25 × 25 paired-end sequencing was performed on an Illumina HiSeq 2500 instrument at the Fred Hutchinson Cancer Research Center Genomics Shared Resource.

For the B cells samples, an alternative library preparation was employed [94].

Sequencing datasets were aligned to the mouse mm10 genome assembly using Bowtie2. Datasets were also aligned to the dmel_r5_51 (*D. melanogaster*) or sc3 (*S. cerevisiae*) assemblies using Bowtie2 depending on the source of the spike-in DNA. Counts per bp were normalized as previously described by multiplying the fraction of mapped reads spanning each position in the genome by genome size [95] or by scaling to spike-in DNA [91]. Peaks were called using MACS. Plots were generated with ngs plot [96], or the R package ggplot2 Motif enrichment was done using HOMER [97].

**Antibodies used.** The following antibodies were used for the indicated techniques with catalog number and supplier listed: anti-DDX4 (WB and immunofluorescence (IF), AB13840, Abcam (Waltham, MA, USA)), anti-Phospho-H3(ser10) (IF, 9706S, Cell Signaling Technology), antibodies against gamma-H2AX (γH2AX) (WB and IF, AB11174, Abcam), anti-Histone H2A (WB, 2578S, Cell Signaling Technology), anti-TXNIP (WB and IF, K0204-3, K0205-3, MBL International), anti-PGK1/2 (WB and IF, SC-28784, Santa Cruz Biotechnology), anti-Ki-67 (IHC, AB16667, Abcam), anti-FAS (WB and IHC, SC-1024, Santa Cruz Biotechnology), anti-TIMP1 (IHC, SC-6832, Santa Cruz Biotechnology), anti-MLX (WB, IHC and ChIP, 85570, Cell Signaling Technology), anti-MLX (IF, 12042-1-AP, Proteintech Group, Inc. (Rosemont, IL, USA)), anti-MondoA (WB and IHC, 13614-1-AP, Proteintech), anti-ChREBP (WB, NB400-135, Novus Biologics (Centennial CO, USA)), anti-MYC (WB, 13987, Cell Signaling Technology), anti-MYC (IHC, SC-764, Santa Cruz Biotechnology), anti-MYCN (WB and IHC, SC-53993, Santa Cruz Biotechnology), anti-MAX (WB, IHC and ChIP, SC-197, Santa Cruz Biotechnology), anti-MNT (WB and ChIP, A303-627A, Bethyl), anti-EOMES (WB, AB23345, Abcam), anti-SOX9 (IF, AB5535, MilliporeSigma (Burlington MA, USA)), anti-OCT4 (WB and IHC, AB184665, Abcam), anti-CPT1A (WB, AB176320, Abcam), anti-IGFBP3 (WB, 10189-2-AP, Proteintech), anti-BIM (WB, 2933S, Cell Signaling Technology), anti-PARP (WB, 9542T, Cell Signaling Technology), anti-SCD (WB, AB19862, Abcam), anti-FASN (WB, AB128856, Abcam), anti-TOMM20 (WB, AB56783, Abcam), anti-ATF4 (WB, 11815S, Cell Signaling Technology), anti-Gamma-Tubulin (WB, T5326, Sigma-Aldrich), anti-Beta-Actin (WB, A5441, Sigma-Aldrich), anti-Mouse HRP (WB, 7076, Cell Signaling Technology), and anti-Rabbit HRP (WB, 7074, Cell Signaling Technology).

**Statistical analysis.** Graphpad Prism was used for all statistical data analysis unrelated to NGS datasets. *p*-Values were calculated by hypergeometric test Student *t* test or 1-way ANOVA with a Dunnett test, when appropriate. For metabolomics analysis, MetaboAnalyst [25] was used. Gene lists of differentially expressed and/or bound targets were analyzed by Enrichr [34] and visualized using Appyter [98]. For RNA-seq and ChIP-Seq analysis, R Studio and Python were used.

## Supporting information

**S1 Fig. (goes with Fig 1). Histological characterization of the WT, HET, and MLX^KO testes and epididymides.** (**A**) Schematic of the targeting construct used to generate deletion of murine *Mlx*. (**B**) Histological analysis of WT, HET, and MLX^KO epididymis stained with hematoxylin and eosin (100×, scale bar = 400 uM). (**C**) Images of cauda epididymal spermatozoa from WT versus MLX^KO mice, the latter with typical abnormal features such as altered tail and head morphology. (**D**) Histological analysis of p51 WT versus MLX^KO testis and epididymis stained with hematoxylin and eosin (100×, scale bar = 400 uM). (**E**) Histological analysis of p51

WT versus MLX$^{KO}$ epididymis stained with hematoxylin and eosin (400×, scale bar = 100 uM). HET, heterozygous; KO, knockout; MLX, MAX-Like protein X; WT, wild-type. (TIF)

**S2 Fig. (goes with Fig 2). Widespread expression of MLX and MondoA transcripts in the human testes.** The Human Protein Atlas database [22] scRNA-seq dataset from adult human testes Guo (2018) [23] was queried for testes expression of (**A**) *MLX* and (**B**) *MLXIP* (encoding MondoA). Both are present in multiple stromal and GC populations, with high expression in primitive GCs and Sertoli (MLX) and Peritubular Myoid cells (MondoA). *Image credit*: *Human Protein Atlas*. Image available from v20.1.proteinatlas.org (http://www.proteinatlas.org). The underlying data for S2A and S2B Fig can be found in S1 Data. GC, germ cell; MLX, MAX-Like protein X; scRNA-seq, single-cell RNA sequencing. (TIF)

**S3 Fig. (goes with Fig 3). Metabolomic data from WT and MLX$^{KO}$ serum.** (**A**) PLS-DA and (**B**) VIP plot from the metabolomic dataset analyzed by MetaboAnalyst 4.0 [25]. (**C**) Heat map of mean-centered serum metabolomics dataset from WT versus MLX$^{KO}$ mice showing the top 20 VIP features from PLS-DA ($N$ = 3). (**D**) A model of altered mitochondrial fuel source based upon serum metabolomics (red indicates up in the MLX$^{KO}$ and green indicates down). Image made with BioRender. (**E**) Sperm count from MondoA WT versus KO mice ($N$ = 5) tested with a paired *t* test. The underlying data for S3A–S3C and S3E Fig can be found in S1 Data. KO, knockout; MLX, MAX-Like protein X; PLS-DA, partial least squares discriminant analysis; VIP, variable importance to projection; WT, wild-type. (TIF)

**S4 Fig. (goes with Fig 4). Comparison of seminiferous tubules and gene expression in WT and MLX$^{KO}$ testes.** (**A**) Histological analysis of testes from 6-month-old males of the indicated genotype stained with hematoxylin and eosin (100×, scale bar = 400 uM). Asterisks mark acellular tubules (**B**) Staining, as in (**A**) demonstrating the range of acellular tubule frequency specific to the whole body MLX deletion. (**C**) Quantification of acellular tubules as the percentage of acellular tubules per 50 tubules (average of 250 total tubules per animal, 4 animals per genotype. Shown is the mean with SEM (* $p < 0.05$, ** $p < 0.01$, *** $p < 0.001$, *** $p < 0.0001$). (**D**) IF staining of WT and MLX$^{KO}$ testes tissue for TXNIP and SOX9. Asterisks marks and acellular tubule. Arrow indicates SOX9+ cells still present in the GC depleted tubule (200×, scale bar = 200 uM). (**E, F**). Human Protein Atlas [22] scRNA-seq dataset from adult human testes Guo 2018 [23] was queried for the established MLX targets: (**E**) *TXNIP* and (**F**) *ARRDC4*. Both are present in multiple stromal and GC populations. *Image credit*: *Human Protein Atlas*. Image available from v20.1.proteinatlas.org (http://www.proteinatlas.org). The underlying data for S4C, S4E, and S4F Fig can be found in S1 Data. GC, germ cell; IF, immunofluorescence; KO, knockout; MLX, MAX-Like protein X; scRNA-seq, single-cell RNA sequencing; WT, wild-type. (TIF)

**S5 Fig. (goes with Fig 5). RNA profiling of testes from WT versus MLX$^{KO}$ mice.** (**A**) CHEA (2016 from Enrichr database) adjusted *p*-values for indicated TF targets associated with up or down in the WT versus the MLX$^{KO}$ RNA-seq data. (**B,C**) WB data from whole testes lysates from WT versus MLX$^{KO}$ mice probed for the indicated proteins. (**D**) IHC analysis of WT versus MLX$^{KO}$ testis and epididymis stained for the proliferation marker Ki-67 (100×, scale bar = 400 uM). Asterisks marks acellular tubule with decreased Ki-67. (**E**) IHC analysis of WT versus MLX$^{KO}$ testis and epididymis stained for TIMP1 (100×, scale bar = 400 uM). (**F**) IHC analysis of WT versus MLX$^{KO}$ testis and epididymis stained for FAS (100× scale bar = 400 uM). CHEA, ChIP set enrichment analysis; IHC, immunohistochemistry; KO, knockout;

MLX, MAX-Like protein X; RNA-seq, RNA sequencing; TF, transcription factor; WB, western blot; WT, wild-type.
(TIF)

**S6 Fig. (goes with [Fig 6]). Molecular, biochemical, and functional validation of GSEA categories from WT versus MLX<sup>KO</sup> mice.** (**A**) WB of cells isolated from the interstitium of WT versus MLX<sup>KO</sup> testes tissue probed for the indicated proteins. (**B**) WB analysis of the WT versus MLX<sup>KO</sup> 3T3 cell lines reconstituted with empty vector or the indicated isoform of MLX probed for the indicated proteins. (**C**) WB analysis of the NTera2 cells treated with the indicated siRNA, probed for the indicated proteins. (**D**) Co-expression analysis of *MLX* and *POU5F1* from the Testicular Germ Cell Tumor Dataset (TCGA PanCancer Atlas). (**E**) Overexpression of the indicated mRNAs from [Oncomine.org] Korkola and colleagues dataset. (**F**) IHC analysis of NTera2 cell xenograft [45] stained for the indicated proteins (200×, scale bar = 200 uM). The underlying data for S6D and S6E Fig and can be found in [S1 Data]. GSEA, gene set enrichment analysis; IHC, immunohistochemistry; KO, knockout; siRNA, small interfering RNA; WB, western blot; WT, wild-type.
(TIF)

**S7 Fig. (goes with [Fig 7]). Enrichment analysis of MLX and MAX binding in the mouse testes.** (**A, A'**) Enrichr analysis of genes bound by MLX (MLX bound) analyzed for enrichment of pathways by MSigDB (**A**) and transcriptional regulators by CHEA (**A'**). (**B, B'**). Enrichr analysis of genes bound by MAX in the presence of MLX, MAX(WT)-Bound analyzed for enrichment of MSigDB (**B**) and CHEA (**B'**). (**C, C'**) Enrichr analysis of genes bound by MAX in the absence of MLX, MAX(MLX<sup>KO</sup>)-Bound analyzed for enrichment of MSigDB (**C**) and CHEA (**C'**). (**D**) Enrichr analysis of genes previously bound by MLX, but only bound by MAX in the absence of MLX, de novo genes MAX(MLX<sup>KO</sup>)-Bound analyzed for enrichment of MSigDB (**D**) and CHEA (**D'**). All Enrichr images created with Appyter [98]. The underlying data for S7A–S7D Fig can be found in [S1 Data]. CHEA, ChIP set enrichment analysis; KO, knockout; MLX, MAX-Like protein X; MSigDB, Molecular Signature Database; WT, wild-type.
(TIF)

**S8 Fig. (goes with [Fig 8]). MLX shares numerous transcriptional targets with MAX and MNT.** (**A**) Venn diagram of the overlap of genes bound by MLX with MAX (Left), and MLX with MNT (Right), in both WT and MLX<sup>KO</sup> 3T3 cells with *p*-value calculated from a hypergeometric test. (**B**) IGV tracks for MLX, MAX, and MNT on the *Arrdc4*, *Fasn*, and *Atf4* promoters from WT and MLX<sup>KO</sup> 3T3 cells. (**C**) Enrichr analysis of the subset of MLX-bound and MAX-bound peaks that are only bound in the MLX<sup>KO</sup> 3T3, de novo MAX sites MAX(MLX<sup>KO</sup>)-Bound for CHEA enrichment. (**D**) Enrichr analysis of MNT-Bound genes only occupied in the presence of MLX, MNT(WT)-Bound for MsigDB enrichment. (**E**) Enrichr analysis of MNT-Bound genes only occupied in the absence of MLX, MNT(MLX<sup>KO</sup>)-Bound for MsigDB enrichment. All Enrichr images created with Appyter [98]. The underlying data for [S8A and S8E Fig] can be found in [S1 Data]. CHEA, ChIP set enrichment analysis; IGV, Integrative Genomics Viewer; KO, knockout; MLX, MAX-Like protein X; WT, wild-type.
(TIF)

**S9 Fig. (goes with [Fig 9]). MLX and MNT regulate metabolism and survival of male GC lines.** (**A**) Relative viable cell number of the GC-2 Spd(ts) cells after siRNA transfection with the indicated siRNA with siDEATH included as a control for siRNA transfection efficacy. (**B**) WB analysis of GC-1-Spg and GC-2 Spd(ts) cells probed for the indicated proteins. (**C**) WB analysis of NTera2 cells transfected with the indicated siRNA and probed for the indicated

proteins. (**D**) Relative viable cell number of the NTera2 cells after siRNA transfection with the indicated siRNA with siDEATH included as a control for siRNA transfection efficacy. Cells were cultured in the presence or absence of 35 uM OA ($N$ = 4 independent experiments, Shown is the mean +/− SEM), $p$-values shown from 1-way ANOVA with a Dunnett test compared with the Control (* $p < 0.05$, ** $p < 0.01$, *** $p < 0.001$, *** $p < 0.0001$). (**E**) Analysis of *MNT* expression from the Human Protein Atlas Single-cell RNA-seq dataset [23] *Image credit*: *Human Protein Atlas*. Image available from v20.1.proteinatlas.org (http://www.proteinatlas.org). (**F, G**) GEO2R analysis of GSE6969 from a published dataset of fertile (normospermic) versus infertile (teratozoospermic) men. LogFC and $p$-values are shown on the table to the right. The underlying data for S9A, S9D, and S9F Fig can be found in S1 Data. GC, germ cell; MLX, MAX-Like protein X; OA, oleic acid; RNA-seq, RNA sequencing; siRNA, small interfering RNA; WB, western blot.
(TIF)

**S1 Data. An Excel file containing the raw numerical data needed to reproduce data in this report with the exception of Fig 3K.**
(XLSX)

**S2 Data. A ZIP file containing the raw.fcs data for the flow cytometry data visualized in Fig 3K in this report.**
(ZIP)

**S1 Raw Images. A PDF file containing the complete original WB images used as sources of WB data depicted in this report.** WB, western blot.
(PDF)

**S1 Table. Metabolomics data.** A table of LFC normalized metabolomics dataset from 3 age-matched littermate WT and MLX<sup>KO</sup> male mice. KO, knockout; LFC, log fold change; MLX, MAX-Like protein X; WT, wild-type.
(CSV)

**S2 Table. RNA-seq DEGs.** A table of DEGs from RNA-seq for both up- and down-regulated transcripts between WT and MLX<sup>KO</sup> testes tissue. DEG, differentially expressed gene; KO, knockout; MLX, MAX-Like protein X; RNA-seq, RNA sequencing; WT, wild-type.
(TXT)

**S3 Table. RNA-seq GSEA.** A table of GSEA of "Hallmarks" from RNA-seq for both up- and down-regulated transcripts between WT and MLX<sup>KO</sup> testes tissue. GSEA, gene set enrichment analysis; KO, knockout; MLX, MAX-Like protein X; RNA-seq, RNA sequencing; WT, wild-type.
(XLSX)

**S4 Table. MLX and MAX targets associated with male fertility in the mouse.** A table showing targets of MLX, MAX, and/or CREM associated with male fertility in the mouse. Shown are gene name, gene function, TF binding, fold change from RNA-seq, and published phenotypes relevant to male fertility. MLX, MAX-Like protein X; RNA-seq, RNA sequencing; TF, transcription factor.
(XLSX)

## Acknowledgments

The authors are grateful to former and current members of the Eisenman Lab who have contributed to the discussion and direction of this project, as well as to the greater community

that comprises the collaborative environment at the Fred Hutch and University of Washington. We also thank the Experimental Histopathology and Genomics shared resources at Fred Hutch.

## Author Contributions

**Conceptualization:** Patrick A. Carroll, Haiwei Gu, James A. Dowdle, Vivek Venkataramani, Vijay Ramani, Donald E. Ayer, Charles H. Muller, Robert N. Eisenman.

**Data curation:** Patrick A. Carroll, Brian W. Freie.

**Formal analysis:** Brian W. Freie, Sivakanthan Kasinathan, Haiwei Gu.

**Funding acquisition:** Daniel Raftery, Donald E. Ayer, Robert N. Eisenman.

**Investigation:** Patrick A. Carroll, Pei Feng Cheng, Haiwei Gu, Theresa Hedrich, Vivek Venkataramani, Xiaoying Wu, Charles H. Muller.

**Methodology:** Sivakanthan Kasinathan, Haiwei Gu, Vivek Venkataramani, Charles H. Muller.

**Project administration:** Robert N. Eisenman.

**Supervision:** Daniel Raftery, Jay Shendure, Donald E. Ayer, Charles H. Muller, Robert N. Eisenman.

**Validation:** Patrick A. Carroll.

**Visualization:** Xiaoying Wu.

**Writing – original draft:** Patrick A. Carroll, Robert N. Eisenman.

**Writing – review & editing:** Patrick A. Carroll, Donald E. Ayer, Robert N. Eisenman.

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
