## [Editor Report · Decision Letter 0]

11 Jan 2021

Dear Dr Eisenman, 

Thank you for submitting your manuscript entitled "MLX balances metabolism and stress to suppress apoptosis and maintain spermatogenesis" for consideration as a Research Article by PLOS Biology. Please accept my apologises for the delay in getting back to you due to the closure of the editorial office over the holiday period. 

Your manuscript has now been evaluated by the PLOS Biology editorial staff as well as by an academic editor with relevant expertise and I am writing to let you know that we would like to send your submission out for external peer review.

Please re-submit your manuscript within two working days, i.e. by Jan 13 2021 11:59PM.

Kind regards,

Richard Hodge, PhD

Associate Editor

PLOS Biology

---

## [Decision Letter · Decision Letter 1]

10 Mar 2021

Dear Dr Eisenman,

Thank you very much for submitting your manuscript "MLX balances metabolism and stress to suppress apoptosis and maintain spermatogenesis" for consideration as a Research Article at PLOS Biology. Your manuscript has been evaluated by the PLOS Biology editors, an Academic Editor with relevant expertise, and by several independent reviewers.

The reviews are attached below. You will see that the reviewers find your conclusions novel and interesting, but they also raise concerns with the degree of insight into the underlying mechanisms that mediate the observed defect in spermatogenesis. Specifically, we note the concern raised by Reviewer #2 regarding the specific cell types that mediate your observed phenotypes. This point has also been raised by the academic editor handling your submission, and we ask that you please include additional experimental data in your revised manuscript that addresses this (such as conditional knockout experiments).

In light of the reviews, we will not be able to accept the current version of the manuscript, but we would welcome re-submission of a much-revised version that takes into account the reviewers' comments. We cannot make any decision about publication until we have seen the revised manuscript and your response to the reviewers' comments. Your revised manuscript is also likely to be sent for further evaluation by the reviewers.

We expect to receive your revised manuscript within 3 months. 

**IMPORTANT - SUBMITTING YOUR REVISION**

*Re-submission Checklist*

*Published Peer Review*

*PLOS Data Policy*

*Blot and Gel Data Policy*

Sincerely,

Richard

Richard Hodge, PhD

Associate Editor, PLOS Biology

rhodge@plos.org

REVIEWS:

Reviewer's Responses to Questions

PLOS authors have the option to publish the peer review history of their article (what does this mean?). If published, this will include your full peer review and any attached files.

Reviewer #1: Yes: Akihiko Okuda

Reviewer #2: No

Reviewer #3: Yes: Ricardo Daniel Moreno Mauro

Reviewer #1 (Akihiko Okuda): In the manuscript, Carroll et al. reported the consequence of loss of MLX in mouse development.

They demonstrate that loss of Mlx does not affect overall viability in mice. However, those male mice are found to be completely infertile, whereas the knockout female mice are able to breed with wild-type and heterozygously knockout male mice.

Their in-depth functional studies revealed that germ cells in Mlx knockout mice are rather defective in the transition from round to elongated spermatids.

Furthermore, their comprehensive gene expression and ChIP-sequence analyses revealed that expression of important metabolic target and apoptosis-related genes are significantly down- and up-regulated in germ cells, respectively, mainly due to the loss of functions of MondoA-MLX and MNT-MLX complexes.

I consider that this is an excellent study that meets the high quality requirement for the journal. However, as described below, there are several portions that should be corrected in the paper.

The authors demonstrated in Figure 5A and Figure S6A, MAX binding sites are extensively altered due to the Mlx knockout in testis and 3T3 cells, respectively. I consider that the authors should conduct systematic bioinformatics analyses to characterize the Mlx-knockout dependent MAX binding sites. Are they enriched for stress and/or metabolism-related genes? Are they enriched for MNT-MLX binding sites? Is Atf4 gene one of such representative example? I assume this because Max signal shown in Figure S6B appears to be higher in MLX knockout than that of wild-type. I am especially interested in gene sets served as physiological binding sites for MLX (53 and 69 sites in testis and 3T3 cells, respectively).

Minor comments:

1 (page 6 line 3 from the bottom) :

It is indeed possible to say that number of spermatozoa is decreased, but it is not possible to judge their mature levels only from the HE staining panels shown in Figure 1G. 

2 (page 7 line 10 from the bottom) :

Ratios of round vs. elongated spermatids cannot be judged with Figure 1H.

If the authors judge the level of normality in morphology by the absence of DDX4 staining, I would recommend to cut and paste the sentences explaining about the meaning of DDX4 staining in epididymis described in page 9 line 11-14 in this portion.

3 (page 9 line 8 and other portions) :

Phosphor-gammaH2AX should be changed to gammaH2AX, since gammaH2AX means phosphorylated H2AX.

Figure 3H, 3I and Figure S2C

Letters along with heat maps cannot read.

In Figure 3 legend:

Panels (A-K) in Figure 3 are appropriated cited in the text, but in Figure legend in which panels E, F and G are explained as a single panel (panel E).

Reviewer #2: Mlx/Mondoa is a heterodimeric basic helix-loop-helix transcription factor with a conserved role in glucose-responsive transcription and regulation of cellular carbohydrate metabolism. However, the physiological roles of these factors in mammals have remained poorly characterized. The manuscript of Carroll and coworkers make a significant advancement on this topic by reporting the role of Mlx and MondoA in mouse male germ cell development. In addition to thorough histological characterization of the phenotype, the study includes extensive genomics and metabolomics datasets that allow the authors to conclude that Mlx regulates genes associated with metabolism, stress and male germ cell development and that MLX loss-of-function alters MYC network transcriptional output. The data is of high relevance to scientists studying Mlx or spermatogenesis. However, for the manuscript to be a strong candidate for PLoS Biology, I would expect to see more advanced and specific main conclusions. The experiments seem generally robust, but issues related to data presentation make it difficult to follow the line of thought and specifics of experimentation. Prior to publication, I request the authors to address the following major concerns:

General importance:

The main finding of the manuscript (the role of Mlx/MondoA in male germ cell development) is novel and interesting, bringing a new dimension to the Mlx/MondoA literature. Moreover, the mouse model presented in this study will be of high importance to the field. The spermatogenesis phenotype is rather thoroughly characterized by histological stainings and specific markers, along with genomic analysis of targets of known regulators of male germ cell development. Moreover, the metabolic role and target genes of Mlx are characterized by state-of-the art metabolomics, genomics and further validated by Western blot analyses. However, despite the wealth of data, the main conclusions of the manuscript (as listed in the 'highlights') remain at a rather general level. The current version of the manuscript does not address sufficiently well, which target genes/cellular processes underlie the observed defects in spermatogenesis or which stage/cell type Mlx/Mondo is necessary for. In my opinion, to be a strong candidate for a journal of broad readership, like Plos Biology, the study should yield more mechanistic insight. 

For example: 

-the rescue observed by oleic acid in teratoma cells serves as a promising starting point for deepening the analysis of the potential the role of Mlx/MondoA regulated lipogenesis in male germ cell development. Are their lipid-related phenotypes observed in the testis of Mlx KO mice or does interference of lipogenesis phenocopy loss of Mlx? 

-immunostaining with Mlx-specific antibodies to analyze Mlx expression/localization in the testis would bring additional insight into the dynamics/cell type specificity of Mlx function.

Presentation:

The manuscript contains massive datasets obtained from various models/sources, some unrelated to testis (such as metabolomics of serum and genomic data from various unrelated cell types, e.g. primary B cells, 3T3 cells). This poses a challenge to coherent data presentation. It would also be helpful to have more explanatory figure legends to eliminate the need jump between results, figures and methods to understand the experimental setup and the main conclusions from the data. Therefore, I strongly encourage the authors to significantly edit the manuscript in a way that better helps the reader to follow the logic of the experimentation and think carefully, if all the data included into the current manuscript is relevant. The authors should make an effort to better walk the reader through the experiments and conclusions, keeping the focus on the main theme of the manuscript (Mlx in male germ cell development). 

Statistics:

The statistical data, indicated by stars, should be displayed in a way that it is clear, which samples are being compared. The statistical test should be mentioned in the Figure legend.

Minor:

Supplemental Figure 7D contains eight 'OE -/+' but only six bars. 

Reviewer #3 (Ricardo Daniel Moreno Mauro): The manuscript entitled "MLX balances metabolism and stress to suppress apoptosis and mantain spermatogenesis" by Carroll et al., gives a nice pack of information about the role of this transcription facor using up-to-date-technology. However, according to the criteria of this reviewer there are some issues that need to be addressed

Results

1.- I think the best controls are heterozygous and not wild type mouse, as the authors use in the manuscript.

2.- Througthout all the manuscript authors use a wide range of different words that induce to confusion. Germ cells is a term generally use to refer spermatogonia, spermatocyte and spermatids. After they leave the testis is better to name them spermatozoa.

3.- Figure 1C-F: Are the same mice in all these figures? Why in some graphs authors show 5 spots (E) while in other 7 (F)?

4.- Figure 1G. Authors shows abcence of spermatozoa in t MLK-KO mice epididymis. Could this be due to a slowdown in epididmis transit from head to cauda? The authors need to examine all the three major segments of the epididmys to rule out this option. 

5.- What the authors mean by "testis and epididymis tissue were disrupted"? Abnormal tissue histology?

6.- Could the authors define what they mean by abnormal and acellular seminiferous tubules?

7.- I suggest a depper morphometric study in these samples (e.g area and diamter of seminiferous tubules)

8.- Authors mention that MLK-KO mice present oligoastheteratozoospermia (OAT). However they do not show that mature cauda epididymal spermatoza have morphological abnormaligies. Their data by flow cytometry only suggest that some round spermatids reach the epididymis, but this is not a proof of tetratozoopermia. Please see these examples ( Mol Hum Reprod . 2021 Jan 22;27(1):gaaa083. doi: 10.1093/molehr/gaaa083; PLoS One. 2013 Nov 28;8(11):e80607. doi: 0.1371/journal.pone.0080607. eCollection 201). On the same token, Figs 2C, D, H show a reduction in %normal motility in MLKKO males. What is normal motility? Asthezoospermia means a reduction in progresive motility.

9.- Figure 3K needs propers antibody controls since the label looks quite non-specific.

10.- Why authors compare their date in mice with samples from teratozoospermic patients, since they suggest that MLK_KO mice are oligoasthenoteratozoospermic?

---

## [Decision Letter · Decision Letter 2]

1 Sep 2021

Dear Dr Eisenman,

Thank you for submitting your revised Research Article entitled "MLX balances metabolism and stress to suppress apoptosis and maintain spermatogenesis" for publication in PLOS Biology. I have now obtained advice from the original reviewers and have discussed their comments with the Academic Editor. 

Based on the reviews, we will probably accept this manuscript for publication, provided you satisfactorily address the remaining points raised by the reviewers. As requested by Reviewer #2, please check the subheading at lines 641 and 642, which should be changed to 'MNT and MLX transcription factors bind metabolic and stress targets shared with MAX'. Please also make sure to address the following data and other policy-related requests:

(A) We would like to suggest the following change to the title of the manuscript, to make it clearer for our broad readership:

"The glucose-sensing transcription factor MLX balances metabolism and stress to suppress apoptosis and maintain spermatogenesis" 

(B) In your ethics statement, please state the method of euthanasia used in the animal studies

(C) You may be aware of the PLOS Data Policy, which requires that all data be made available without restriction: http://journals.plos.org/plosbiology/s/data-availability. For more information, please also see this editorial: http://dx.doi.org/10.1371/journal.pbio.1001797

Regardless of the method selected, please ensure that you provide the individual numerical values that underlie the summary data displayed in the following Figures, as they are essential for readers to assess your analysis and to reproduce it:

Figure 1B-F, 3A-H, 3K, 4B-K, 5A-C, 5G-I, 6E, 6G-H, 7A, 7C-H, 8A-C 9A-B, 9D, 9H, Figure S2A-B, S3A-C, S3E, S4C, S4E-F, S6D-E, S7A-D, S8A-E, S9A, S9D-F.

(D) Please also ensure that each of the relevant figure legends in your manuscript include information on *WHERE THE UNDERLYING DATA CAN BE FOUND*, and ensure your supplemental data file/s has a legend

(E) Please ensure that your Data Statement in the submission system accurately describes where your data can be found and is in final format, as it will be published as written there. Please remove the names of the specific individual datasets contained within GSE165820 and state that the underlying data for the RNA-seq and ChIP-seq datasets can be found at GSE165820.

(F) We require the original, uncropped and minimally adjusted images supporting all blot and gel results reported in the following Figures:

Fig 1A, 6A-D, 8E, 9C, 9E-G, Fig S5B-C, S6A-C, S9B-C. 

We will require these files before a manuscript can be accepted so please prepare and upload them now. Please carefully read our guidelines for how to prepare and upload this data: https://journals.plos.org/plosbiology/s/figures#loc-blot-and-gel-reporting-requirements

We expect to receive your revised manuscript within two weeks. 

*Published Peer Review History*

*Early Version*

Sincerely,

Richard

Richard Hodge, PhD

Associate Editor, PLOS Biology

rhodge@plos.org

Reviewer remarks:

Reviewer #1:

The authors addressed all my concerns appropriately and I have not any further comments other than recommending the publication of this manuscript in PLoS Biology.

Reviewer #2:

The authors have addressed my original concerns very thoroughly and the manuscript is significantly improved. I am happy to recommend it to be accepted to Plos Biology. Before publication, please check the subheading "MNT and MAX transcription factors bind metabolic and stress targets shared with MAX" (lines 641 and 642).

I wish to congratulate the authors for this groundbreaking study.

Reviewer #3 (Ricardo D. Moreno, signs his review):

I think the authors have made a great work. I have no further questions.

---

## [Editor Report · Decision Letter 3]

24 Sep 2021

Dear Bob,

On behalf of my colleagues and the Academic Editor, Masahito Ikawa, I am pleased to say that we can in principle offer to publish your Research Article "The glucose-sensing transcription factor MLX balances metabolism and stress to suppress apoptosis and maintain spermatogenesis" in PLOS Biology, provided you address any remaining formatting and reporting issues. These will be detailed in an email that will follow this letter and that you will usually receive within 2-3 business days, during which time no action is required from you. Please note that we will not be able to formally accept your manuscript and schedule it for publication until you have made the required changes.

PRESS

Sincerely, 

Richard

Richard Hodge, PhD

Associate Editor, PLOS Biology

rhodge@plos.org

PLOS
